# Research

evolution, theoretical biology

cultural transmission, cultural evolution, cultural linkage, cultural hitchhiking

**Author for correspondence:**
D. Justin Yeh
e-mail: justin_yeh@eva.mpg.de

# Cultural linkage: the influence of package transmission on cultural dynamics

D. Justin Yeh, Laurel Fogarty and Anne Kandler

Department of Human Behavior, Ecology and Culture, Max Planck Institute for Evolutionary Anthropology, Leipzig, Germany

DJY, 0000-0003-2463-9469; LF, 0000-0002-1031-3786; AK, 0000-0003-3766-6597

Many cultural traits are not transmitted independently, but together as a package. This can happen because, for example, media may store information together making it more likely to be transmitted together, or through cognitive mechanisms such as causal reasoning. Evolutionary biology suggests that physical linkage of genes (being on the same chromosome) allows neutral and maladaptive genes to spread by hitchhiking on adaptive genes, while the pairwise difference between neutral genes is unaffected. Whether packaging may lead to similar dynamics in cultural evolution is unclear. To understand the effect of cultural packages on cultural evolutionary dynamics, we built an agent-based simulation that allows links to form and break between cultural traits. During transmission, one trait and others that are directly or indirectly connected to it are transmitted together in a package. We compare variation in cultural traits between different rates of link formation and breakage and find that an intermediate frequency of links can lower cultural diversity, which can be misinterpreted as a signature of payoff bias or conformity. Further, cultural hitchhiking can occur when links are common.

## 1. Introduction

Defining and quantifying the complexity of a cultural trait is a notoriously difficult task in archaeology and anthropology. Many definitions of complexity rely on some quantification of an artefact's component parts and the extent to which those parts are integrated (e.g. [1,2]). Similarly, Arthur ([3], p. 28) offers a definition of 'technology' as 'an assemblage of practices and components' that, together, serve a purpose. This definition suggests that strong links between once-independent components may be a fundamental feature of complex human technology ([3], pp. 35–37). The formation and maintenance of such links might change how complex technological traits are transmitted and, importantly, how they evolve. Complex human technology represents an extreme of stable, linked cultural packages. However, before we can attempt to understand technology from this point of view, we must first establish the fundamentals: how do links between cultural traits emerge? How are they transmitted? And how should the existence of links change our expectations for the outcome of cultural evolutionary processes?

In this paper, we focus on the last question and present a model where cultural traits can be linked and transmitted as a package. Linkage between individual evolving components is not a new concept and the analogous genetic mechanism, genetic linkage, is well studied. When chromosomes are inherited, genes on the same chromosome are transmitted together unless recombination occurs, in which case a segment of the chromosome is replaced with the corresponding segment from another chromosome. Genetic linkage has many profound evolutionary consequences, such as allowing neutral or deleterious alleles to spread by hitchhiking (e.g. [4]), hindering adaptation (e.g. [5]), increasing mutation load (e.g. [6]) and speciation (e.g. [7]). The strong effects of linkage on genetic evolution hint that links between cultural traits might change evolutionary dynamics

in unexpected ways. However, several differences between cultural and genetic transmission limit our ability to predict the effects of cultural linkage from models of genetic linkage. For example, while genes are arranged side by side on a chromosome, there exists no 'cultural chromosome'. In the most general scenario, any cultural trait could be linked to any other producing clusters of traits tightly linked in a network and difficult to separate.

Many have tried to infer underlying cultural transmission processes from population-level patterns [8–12], including the tendency for certain cultural variants to co-occur especially in cross-cultural comparison [13–18]. Akin to the relationship between genetic linkage disequilibrium and genetic linkage, this 'cultural packaging' is a population-level phenomenon that can, but need not, be the result of co-transmission. The existence of a persistent cultural package may be owing to, for example, population structure [13], synergistic effects [13,19] or co-transmission as we show below.

Existing models of cultural evolution often treat traits as completely packaged together, or independent units that can be studied in isolation. As interest increases in models that involve multiple traits such as cultural niche construction, cumulative culture, cultural phylogeny and correlated evolution, there is an increasing need for models that incorporate an appropriate theory of cultural linkage. A complete theory of cultural linkage should include (i) a solid psychological theory of link formation and maintenance, (ii) a description of the transmission of linked traits and the links between them, and (iii) a model of how innovation interacts with linked traits and how novel traits arrive in cultural packages.

In the following, for each of these three essential components, we make the simplest possible assumptions. However, a deeper understanding of link formation mechanisms will be crucial to understanding the role of linkage in any real cultural system and should be the focus of urgent future work. Initially, we assume traits have no synergistic effects (i.e. certain combination of traits are not functionally better or worse) nor are any incompatible. The links form randomly at a fixed rate between any two traits in an individual's cultural repertoire. This is consistent with a mechanism of link formation where, for example, a role model demonstrates actions in sequence. An individual learning those actions might learn both the actions and the sequence—in other words, the actions and the temporal links between them. Of course, a huge diversity of mechanisms is likely to exist. For example, traits may be transmitted together owing to causal reasoning [20], prestige-biased transmission [21] or at the level of trait structure, many complex cultural objects consist of smaller components, perhaps physically linked, which may be transmitted together.

The aim of this paper is to understand whether links between cultural traits alter the cultural composition of the population compared to independent trait transmission. To this end, we model the evolution of cultural traits under various cultural transmission processes and investigate how the population-level signature of such processes, summarized by the level of cultural diversity, changes if we allow links to exist between cultural traits. This analysis provides us with first results on how the existence of links between cultural traits can interfere with signatures of cultural evolution and our ability to detect cultural transmission processes from cultural frequency data. We discuss the circumstances under which we expect linked cultural transmission to be important—where we need to include a realistic

theory of cultural linkage in order to replicate the dynamic of cultural change—and the circumstances under which current theory is sufficient.

## 2. Simulation framework

Below we provide model details and all information required by the Overview, Design concepts, and Details protocol for simulation studies of this type [22]. We develop a model simulating cultural change in a population where each individual carries a number of cultural traits and links between those traits. Individuals synchronously choose another individual with whom they interact (their interaction partner). They can then change their cultural traits and the links between them through horizontal transmission of cultural packages from their interaction partner. Links can transmit, break and form, and the traits can change through random innovation. A model with static and non-transmissible links between traits is also outlined in the electronic supplementary material, appendix S1.

We consider a population of $N$ individuals, possessing $h$ cultural traits of which each can take $k$ possible variants. As described below, each individual can form a link between any two traits, which influences the transmission dynamic. We do not assume any demographic processes: individuals are not lost or added to the population.

### (a) Cultural traits and their pay-offs

Each individual has a pay-off value which is the product of the pay-off effects of all variants adopted. Examples of pay-off may be forage success, monetary income or gifts received. If all variants of a trait provide the same pay-off, we say the trait is neutral. By contrast, if the variants of the trait provide different pay-offs, we label the trait 'functional' (following the terminology of [23]). We assume that, where relevant, variant 1 gives the highest pay-off and variant $k$ the lowest. The pay-off of variant $j$ of trait $i$ is

$$f_{i,j} = 1 - \frac{j-1}{k-1} s_i, \tag{2.1}$$

where $s_i \in [0, 1]$ describes the maximum pay-off difference between variants at trait $i$, thus a neutral trait always has $s_i = 0$. The pay-off of individual $r$ is

$$f^r = \prod_{i=1}^{h} f_{i,j(r)}, \tag{2.2}$$

where $j(r) \in \{1, 2, \ldots, k\}$ denotes the variant adopted by individual $r$ for trait $i$.

For convenience, we refer to variant $k$ in functional traits as a 'detrimental' variant in the sense that, other traits being the same, an individual carrying variant $k$ always has a lower pay-off than those carrying another variant. We make no assumption about whether the variant causes direct harm to the individual.

### (b) Cultural transmission dynamics

At the beginning of each timestep, all individuals simultaneously choose an interaction partner other than themselves with probability $p^r$ according to either unbiased, pay-off-biased, or conformist-biased transmission. In the unbiased case, each individual, $r$, is chosen as an interaction

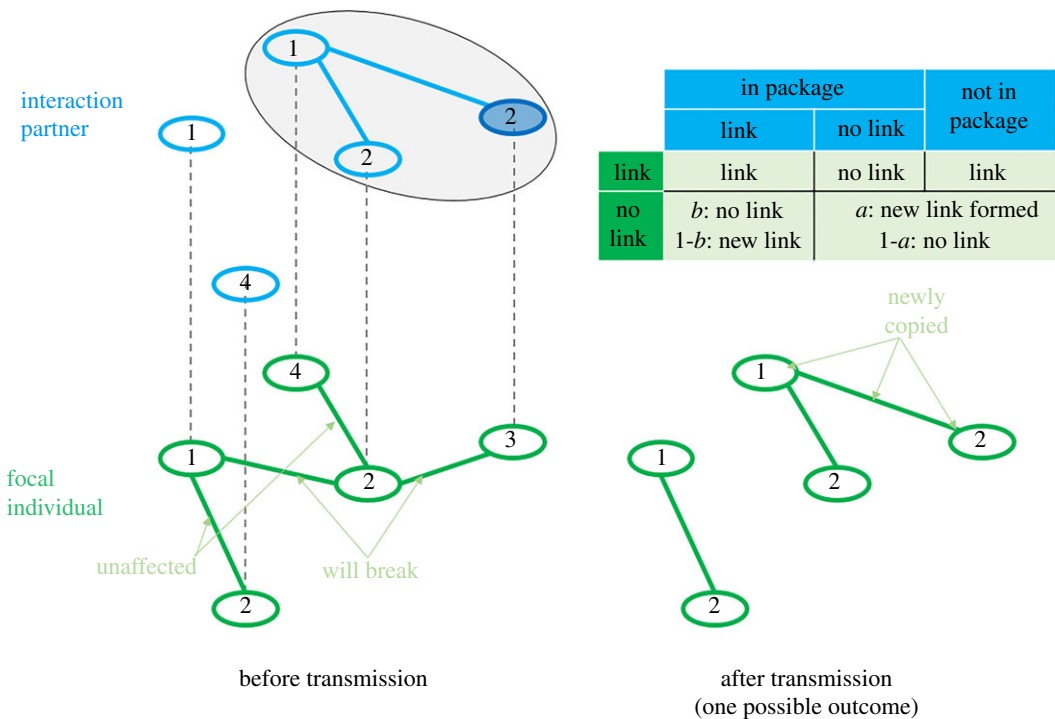

| | in package | | not in package |
|---|---|---|---|
| | link | no link | |
| link | link | no link | link |
| no link | $b$: no link<br>$1-b$: new link | $a$: new link formed<br>$1-a$: no link | |

**Figure 1.** An example of the dynamic of cultural transmission. The small ellipses represent traits, numbers indicate the variants at those traits and straight lines indicate links. The focal individual chooses a trait to learn (dark ellipse) and everything connected to it forms the package to be transmitted (large ellipse). The table on the upper right summarizes the outcomes of link transmission. Traits and links completely outside the package are unaffected. The focal individual attempts to copy all traits in the package, each attempt is successful with probability $c$. Within the package, an existing link in the focal individual is unaffected when the interaction partner also has this link, otherwise it is broken. Where the focal individual has no link, but the interaction partner has, the link is copied at rate $1-b$. The focal individual loses all links between traits inside the package and those outside. (Online version in colour.)

partner with probability $p^r = 1/(N-1)$. In the pay-off-biased case, the probability is proportional to the pay-off, $f^r$, so that $p^r = f^r / \sum_r f^r$. In the conformist-biased case, $p^r = (1 - (1 - q_r)s_\kappa) / \sum_r (1 - (1 - q_r)s_\kappa)$, where $q_r$ is the frequency of the variant carried by $r$ and $s_\kappa$ is the strength of conformity [21]. We assume that the frequency of each trait variant and the total pay-off $f^r$ of each individual are observable, while the effect of each single variant is not. Multiple individuals may choose the same interaction partner.

After this process, the focal individual randomly picks a cultural trait to copy. The interaction partner demonstrates its variant of the chosen cultural trait and the variants of all traits linked to the chosen trait (directly or indirectly, figure 1). The focal individual attempts to copy all variants in this package. As individuals are blind to the effects of single variants, the focal individual may copy a variant of lower pay-off. Each variant is copied successfully with probability $c$, while unsuccessful copying events mean that the focal individual keeps its original variant. When $c < 1$, larger packages become harder to learn than smaller packages. Furthermore, the focal individual acquires each link in the package with probability $1-b$ (with probability $b$ a link in the package is broken). The focal individual loses links between traits inside the package that the interaction partner does not have, and links between traits inside the package and those outside (figure 1).

After transmission, new links randomly form at the rate of association, $a$, between any of an individual's traits that are unlinked. For each trait, with probability $\mu$, individuals then randomly innovate by switching to another variant. Here, we define innovation *sensu* Cavalli-Sforza & Feldman [24] as a random and undirected change. The transition rate

between any pair of variants is $\mu/(k-1)$. Electronic supplementary material, appendix table SA2 provides a summary of all model parameters.

## (c) Simulation setup

At the beginning of a simulation, each individual is randomly assigned a variant at each trait and no links between traits exist. In each timestep, the dynamics described above are iterated, leading to changes in the frequencies of the different variants of the cultural traits. A single simulation run consists of a burn-in period of 5000 timesteps where the transmission process is unbiased followed by an additional 2000 timesteps under unbiased or pay-off-biased transmission. For conformist-biased transmission, we model only one trait and thus no burn-in is needed. At the end of each timestep, variant frequencies are recorded.

In the pay-off-biased scenario, we are interested in cultural hitchhiking. To more clearly see the effects of hitchhiking when it occurs we make some adjustments to the frequency distribution of traits 1 and 2 at the end of the burn-in period. The adjustments represent the most challenging scenario for a hitchhiking trait and are, therefore, conservative. We adjust the following: for trait 1, $N-1$ individuals are assumed to carry variant 4 (the lowest pay-off variant, see equation (2.1)) and a single individual (innovator) carries variant 1 (the highest pay-off variant). For trait 2, we reset the variant numbers so that the innovator has the rarest, and lowest pay-off, variant 4 (which we label 'the associated variant'). We focus on whether the frequency of the associated variant can increase by hitchhiking on the high pay-off variant carried by the innovator.

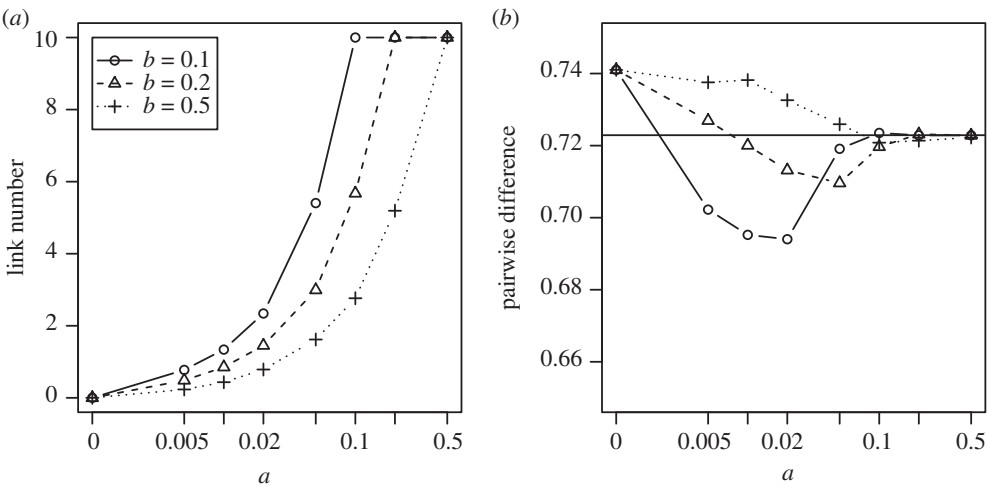

**Figure 2.** Mean frequency of (*a*) links in the population and (*b*) pairwise difference of trait 1 at equilibrium with varying *a* and *b*. Horizontal lines mark the Wright–Fisher expectation $\pi^{WF}$ for a single unlinked trait. Figure 4 provides an idea of how wide the distribution is for one data point. Note: the zero on the log-scaled horizontal axis is placed at where $10^{-4}$ would be. $c = 0.99$, $N = 1000$, $h = 5$, $k = 4$, $\mu = 0.01$.

For each parameter combination, we run 700 simulations. The Matlab code for the simulation and R code for the analysis are available on the Dryad Digital Repository [25].

## 3. Analysis

We track the frequencies of variants of cultural traits over time under different transmission scenarios. To summarize the cultural composition of the population, we calculate the pairwise difference, denoted $\pi_i$, for each cultural trait *i*:

$$\pi_i = 1 - \sum_{j=1}^{k} \frac{q_{i,j}(q_{i,j} - 1)}{N(N - 1)} , \qquad (3.1)$$

where $q_{i,j}$ is the frequency of variant *j* at trait *i*. The pairwise difference $\pi_i$ describes the probability that two randomly selected individuals have different variants at trait *i*. It is also referred to as heterogeneity index and very closely related to measures of diversity such as the Simpson index. If $\pi_i$ is low, the population is homogeneous with respect to trait *i*, i.e. most individuals carry the same variant. By contrast, if $\pi_i$ is high, the variants occur with similar frequencies in the population.

We aim to understand whether links between cultural traits alter the cultural composition of the population compared with the situation where traits evolve independently using three scenarios. First, we consider a neutral system where the cultural transmission dynamic is unbiased. The aim of this analysis is to explore whether the presence and transmission of links can obscure the signature of unbiased cultural transmission (as expected under standard neutral models such as the Wright–Fisher model). Second, we assume that trait 1 is functional ($s_1 = 0.9$), but all other traits remain neutral. The transmission process is now pay-off-biased with the pay-off of an individual determined by its trait 1 variant (equation (2.1)). The aim here is to understand whether the presence of links can cause a neutral trait to hitch-hike on a functional trait and whether the signature of the neutral trait may therefore resemble one under pay-off-biased transmission. Third, we assume that traits 1 and 2 are functional ($s_1 = 0.9$, $s_2 \in \{0.1, 0.2\}$) but all other traits remain neutral. As above, the transmission process is pay-off-biased

and the pay-off of an individual is determined by its trait 1 and 2 variants. The aim of this analysis is to see whether the presence of links can cause a detrimental variant to hitchhike on trait 1 and lead to suboptimal adaptation of the population.

### (a) Scenario I: neutral system

We start our analysis by comparing mean pairwise differences $\pi_i$ at the end of independent simulations for various values of *a* and *b* to the mean value expected under a standard Wright–Fisher model $\pi^{WF} \approx 2N\mu(k - 1)/(k - 1 + 2N\mu k)$ (e.g. [26]). We find that the pairwise difference, and therefore, the level of cultural diversity becomes lower when links exist (figure 2*b* and compare data points at $a = 0$, indicating a situation of no links in the cultural system, with higher values of the association rate *a*). Two important processes contribute to this effect.

First, we note that, in the absence of other forces, trait transmission acts to decrease pairwise difference and innovation acts to increase it. Links change the relative contribution of these two effects. More links mean that packages are larger and more trait variants are transmitted each round. Because the rate of innovation is independent of the transmission process, it remains constant while the effect of transmission increases. To show this more clearly, we ran a simulation in which the focal individual randomly picked *n* traits to copy irrespective of links. When $n = 0$, there is no transmission, and so the pairwise difference is maximized. As the number of traits being copied in each timestep increases, the pairwise difference decreases. When $n = 5$, all trait variants are transmitted and they can be viewed as one. We compared these results to the pairwise distance we expect from a Wright–Fisher model with one trait. To do this, we scale the rate of innovation by $h/nc$ to account for the fact that innovation occurs at a constant rate in each timestep, while the traits are, on average, only successfully transmitted every $h/nc$ timesteps. This accounts for the difference between an unlinked five-trait case ($n = 1$) and the fully linked case ($n = 5$). Both the Wright–Fisher expectation (dashed line) and the simulation results (circles) are shown in figure 3*a* (these are similar to results with static links, electronic supplementary material, appendix S1). Manipulating the package size in this way does not account for the full extent of the decrease in pairwise difference seen in figure 3*b*,

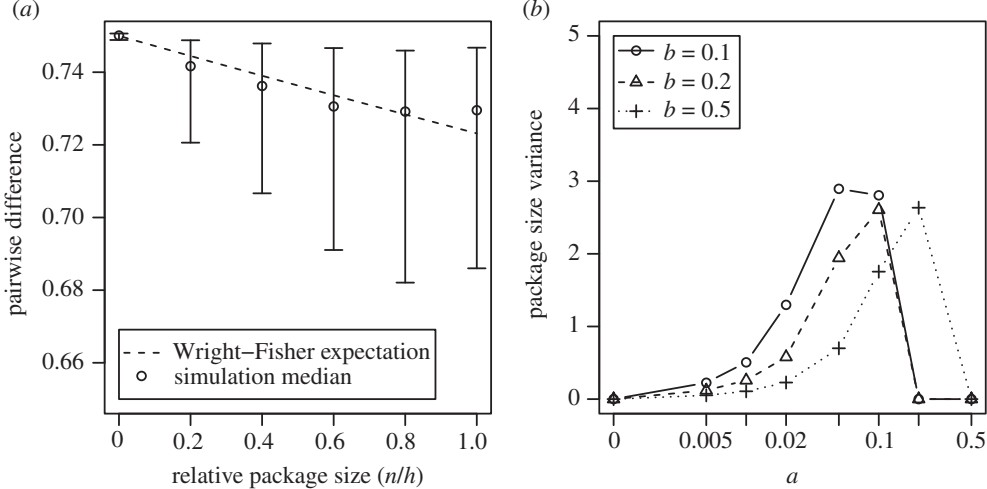

**Figure 3.** Mechanisms that allow links to lower pairwise difference. (a) Drift is more effective when package size is large. Here, we show medians and 90% intervals for simulations with no links but allowing different package sizes. With larger package, pairwise differences become lower. The dashed line is the expectation under the Wright–Fisher model after scaling the mutation rate by the relative package size and the copy rate. (b) The variance of package size is higher at intermediate link frequency, favouring variants in large packages. $c = 0.99$, $N = 1000$, $h = 5$, $k = 4$, $\mu = 0.01$.

especially when the frequency of links is at an intermediate level (e.g. $a = 0.05$, $b = 0.1$), indicating that there is a second process through which links affect cultural diversity.

This second process involves the effect of variable package size. In general, links between cultural traits increase the probability that those traits are transmitted, because the probability that a package is chosen to be copied is proportional to the relative size of the package: variants that happen to be in larger packages have a higher rate of transmission. In figure 3b, we show the variance of the package size—where the variance is high the benefit of being in a large package is also high. Variance can be low when all traits are linked (i.e. when $a$ is high and $b$ is low) or when no traits are linked (i.e. when $a$ is low). Where all traits are linked, they can be viewed as one single trait and there is no longer a difference in transmission rate between variants. This can be seen again in figure 2b where pairwise difference falls as links become more common and then rises again when $a$ is very large. Interestingly, the pairwise difference gets higher when $b$ is high even if the link frequencies are similar because any packages that form are broken down very quickly, so that no single variant is consistently in a large package over an extended period of time. This is similarly true in a model with static links (electronic supplementary material, appendix S1). These results suggest an important interplay between package size and package stability over time in driving cultural evolution with linked traits.

Next, we explore the possibility of rejecting unbiased transmission as the probable underlying transmission process if we falsely assumed no links between cultural traits. Following [27], we ran 1000 simulations for certain parameters and calculated areas of overlap between the distributions of final pairwise differences $\pi_1$ under unbiased linked transmissions with $a = 0.01$, $b = 0.1$ and unbiased unlinked transmission with $a = 0$ as well as the standard Wright–Fisher model. The areas of overlap represent a general measure of equifinality. (If we have an empirical estimate of the pairwise difference of the cultural system under consideration, we can make more precise claims.) Values of the area of overlap close to 0 mean that the presence of links between some traits will lead to very different values of $\pi_1$ compared with unlinked transmission. High values indicate that the presence of links between traits has almost no

influence on $\pi_1$. It is clear that the presence of links moves the pairwise difference distributions towards lower pairwise difference values (figure 4a). The area of overlap between linked and unlinked unbiased transmission is 0.25 and between linked unbiased transmission and the Wright–Fisher model is 0.61. The electronic supplementary material, appendix S1 contains the results with static links and electronic supplementary material, appendix S3 contains the results with different $c, \mu, k, h$ and $N$. Consequently, if we analyse a cultural system where package transmission is possible with statistics that assume independent trait transmission, we may wrongly interpret low pairwise difference values as evidence against unbiased transmission.

We have seen that the existence of links leads to lower pairwise difference values, and therefore levels of cultural diversity. But it is well known that other cultural transmission processes such as conformist-biased transmission, prestige-biased transmission, pay-off-biased transmission or indirectly biased transmission result in relatively low diversity [21,28–30]. Following the same logic as above, we ask whether we can mistake unbiased linked transmission for (unlinked) pay-off-biased or conformist-biased transmission. We calculate the areas of overlap between unbiased, linked transmissions with $a = 0.01$, $b = 0.1$ and unlinked pay-off-biased transmission ($s_1 = 0.07$, $s_i = 0$, $i = 2, \ldots, 5$) and conformist-biased transmission ($s_\kappa = 0.03$). The areas of overlap (figure 4b) between unbiased linked and pay-off-biased unlinked transmission is 0.79, and between unbiased linked transmission and conformist-biased transmission is 0.91. These relatively large values point to the problem of equifinality: a large range of pairwise difference values can be obtained by either of these three transmission processes and consequently lower pairwise difference values could be interpreted as evidence for pay-off-biased or conformist-biased transmission while in fact they are caused by the existence of links in the cultural system.

Summarizing, in a neutral system, where all cultural traits are transmitted without bias, links affect the cultural composition of the population. In situations of low (but positive) or intermediate link frequency, the level of cultural diversity as measured by the pairwise difference can be lower than the

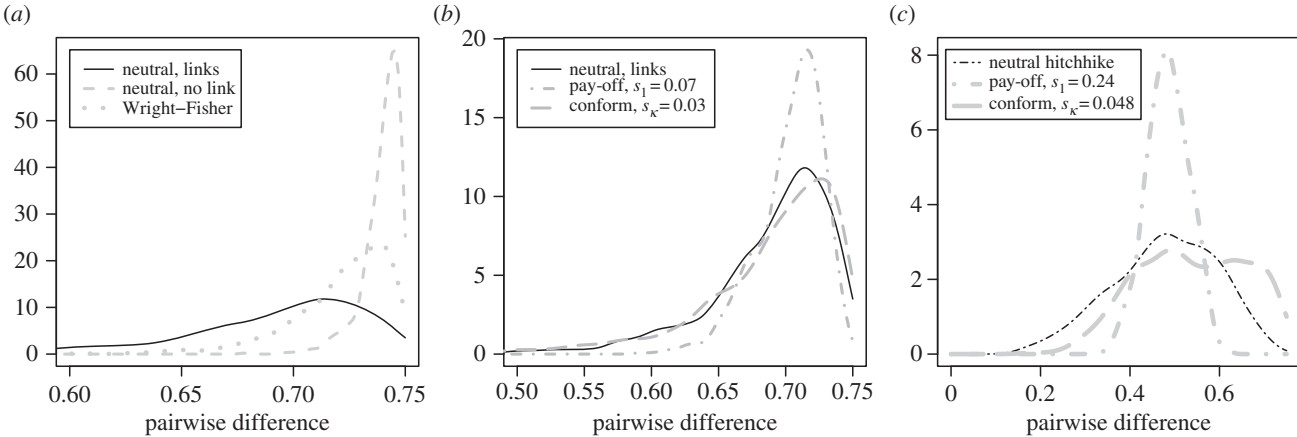

**Figure 4.** Distributions of pairwise difference under (*a*) unbiased transmission with different linkage conditions, (*b*) linked unbiased, unlinked pay-off-biased and conformist-biased transmission, and (*c*) linked pay-off-biased (but shown for the neutral associated variant), unlinked pay-off-biased and conformist-biased transmission. Unsurprisingly, when there are no links or when everything is in a stable package, neutral traits have high pairwise difference at equilibrium. However, when links can form and break ($a = 0.01$, $b = 0.1$), neutral traits have lower pairwise difference values that can be difficult to distinguish from a trait that is transmitted under pay-off or conformity bias. For cases with pay-off-biased transmission, $s_2 = 0$. $c = 0.99$, $N = 1000$, $h = 5$, $k = 4$, $\mu = 0.01$.

Wright–Fisher expectation (figure 2). If, however, link frequency in the population increases further so that all cultural traits usually transmit together, i.e. the package composition is stable, we observe no differences to the standard Wright–Fisher model. In some cases, the existence of links may have serious implications for inferring underlying processes of cultural transmission if data are analysed with methods that are based on the assumption of independent trait transmission.

## (b) Scenario II: neutral trait hitchhiking on a functional trait

To investigate the possibility of cultural hitchhiking when one trait is under pay-off-biased transmission, we find the end of the cultural sweep, defined as the earliest timestep in which the frequency of the highest pay-off variant 1 of trait 1 does not increase. We then calculate the proportion of simulations in which the frequency of the associated variant of trait 2 at this timestep exceeds the absolute majority (i.e. frequency = 0.5). For the simulations in which a majority is achieved, we determine the mean time for which the frequency of the associated variant stays above 0.5 after the cultural sweep.

The first column of figure 5 shows that a neutral variant associated with a trait sweeping to high frequency (typically around 0.9) can reach a high frequency through hitchhiking in a large proportion of simulations. After the sweep, drift and innovation will drive the associated variant to its lower equilibrium frequency. Interestingly, links have two opposing effects on the time the associated variant can stay at a high frequency. First, as explained in figure 3*a*, links can increase transmission, which shortens the time the associated variant can stay at a high frequency. Second, links lead to a higher frequency at the end of the sweep (figure 5*b*), which lengthens this time (figure 5*c*). We show only estimates of this mean time for parameter constellations where hitchhiking occurred in over 5% of simulations and observe that an intermediate or high link frequency does not influence the mean time at high frequency for neutral hitchhiking traits greatly (figure 5*c*).

To explore whether a neutral associated trait can be misidentified as being subject to pay-off- or conformist-biased transmission, we determine the pairwise difference

distribution of trait 2 at the end of the selective sweep (again with $a = 0.01$, $b = 0.1$) and calculate the area of overlap between the pairwise difference distributions of an unlinked trait under pay-off-biased transmission with $s_1 = 0.24$ at the end of a simulation, i.e. at equilibrium (figure 4*c*). The area has a value of 0.58 but all pairwise difference values generated under pay-off-biased transmission (grey dashed-dotted line) can also be generated by the neutral associated trait (black dashed-dotted line). Further, the area of overlap between the pairwise difference distributions of the neutral associated trait and an unlinked trait under weak conformist-biased transmission with $s_\kappa = 0.048$ has a value of 0.83, pointing again to the problem of equifinality.

In summary, the existence of links facilitates hitchhiking between functional and neutral cultural traits. During the sweep of the functional trait, the pairwise difference of the hitchhiking neutral trait decreases rapidly as one variant reaches high frequency. It may take some time after the sweep for the hitchhiking trait to return to the expectation of unbiased linked transmission. In other words, a neutral hitchhiking trait can possess the signature of a functional trait in the sweep phase and for some time after the sweep. Additionally, links between cultural traits increase the rate at which the highest pay-off variant of trait 1 spreads, because links increase the size of the transmitted package (electronic supplementary material, appendix S4).

## (c) Scenario III: hitchhiking between two functional traits

Following the procedure outlined above, we calculate the proportion of simulations in which the frequency of a detrimental associated variant exceeds absolute majority at the end of a sweep, as well as the mean time in which that majority can be maintained.

Figure 5*d–i* shows that hitchhiking of detrimental associated variants can happen, albeit less often (figure 5*d,g*) compared to the situation of a neutral trait shown in the first column, and it ends at a lower frequency (figure 5*e,h*). We show in figure 5*e,h* that hitchhiking becomes less likely the more detrimental the associated variant becomes. Further, the mean time the associated variant stays at high frequency

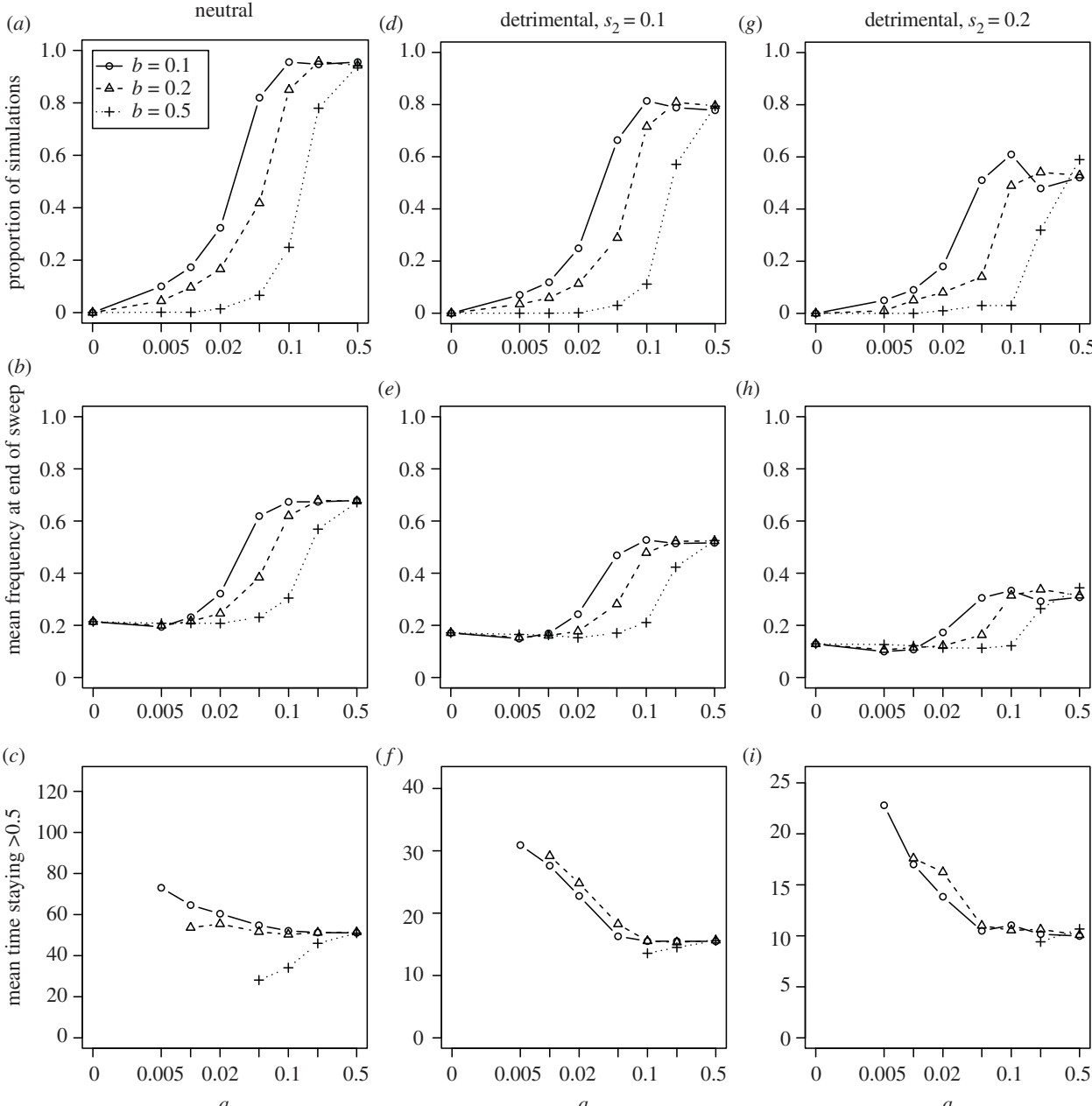

**Figure 5.** (a–i) A neutral or detrimental variant at trait 2 can hitchhike on a beneficial variant at trait 1. The first row shows the proportion of simulations in which a rare variant at trait 2 can reach absolute majority before the highest pay-off variant of trait 1 reaches its maximum. The second row shows the mean frequency of the associated variant at the end of the sweep, which increases with link frequency. The third row shows, among simulations in which the associated variant can reach a frequency greater than 0.5, the mean time in which the majority is maintained after the end of the cultural sweep. $c = 0.99$, $N = 1000$, $h = 5$, $k = 4$, $\mu = 0.01$.

is much shorter than in the neutral case. After the cultural sweep, almost the whole population has adopted variant 1 for trait 1 and consequently the differences in pay-off between the individuals are mostly caused by trait 2 (see equation (2.2)). Eventually, variant 1 of trait 2 will arise and its frequency will increase. This process occurs faster when there are more links, as a high link frequency increases the chance that trait 2 is included in the transmitted package leading to a smaller mean time that the associated variant stays at high frequency (figure 5f,i). More links drive the associated variant to higher frequencies after the sweep but also contribute to the rapid replacement of detrimental variants by less detrimental innovations.

In summary, links in a cultural system can lead to the spread of a detrimental variant, and temporarily suboptimal adaptation of the population. However, after a cultural sweep, the detrimental variant decreases quickly in frequency while the

population reaches its optimum. Further, hitchhiking becomes less likely the more detrimental the associated variant.

## 4. Discussion

We explored whether links between cultural traits alter the cultural composition of a population compared to a population with independently evolving traits. In this model, links between traits mean that those traits are transmitted together as a package. Considering different modes of cultural transmission and link frequencies, we recorded the cultural composition of the population using pairwise difference, a measure of cultural diversity. By design, the modelling assumptions are as simple as they can be to begin to understand the effect of links on cultural evolution. In particular, our model assumes that links form and break at random at a fixed rate independent of the cultural traits, remaining agnostic as to

the origins of links. Whether these assumptions are appropriate will depend on the study system and the nature of specific links.

## (a) The influence of links on the cultural evolutionary dynamic

We found that the existence of links between cultural traits, especially when the link frequency is low or intermediate and individuals can transmit packages of varying sizes, can lead to lower pairwise difference compared to the Wright–Fisher expectation under unbiased transmission. If link frequency is high enough to link most cultural traits then the package can be viewed as a single cultural trait and the pairwise difference matches the Wright–Fisher expectation. The relationship of this model to the Wright–Fisher expectation is robust to changes in $\mu$, $k$, $h$ and $N$ (electronic supplementary material, appendix S3).

Links between cultural traits can facilitate hitchhiking of a neutral trait on a 'functional' one, and between two functional traits leading to the spread of a detrimental variant through the population (with 'detrimental' defined above). In other words, the existence of links can lead to a situation where hitchhiking may cause a neutral trait to resemble a trait under pay-off-biased transmission in cultural frequency data. Hitchhiking can lead to suboptimal adaptation of the population as a detrimental variant reaches relatively high frequencies. However, intuitively, if the pay-off difference between the best and worst variant is too high, this dynamic breaks down and the detrimental variant cannot spread.

## (b) Comparison between cultural and genetic evolution

Under the neutral model of genetic evolution, linkage does not affect the expected pairwise difference [31,32]. The reason an effect is present in cultural evolution is that the number of cultural traits transmitted is variable, unlike in the case of genes. In our model, the linkage patterns can vary between individuals and are copied along with trait variants, allowing variants in large packages to spread faster than those in smaller ones. These differences between cultural and genetic evolution disappear if the link frequency in the population is high. In this situation, almost all traits are linked, and the entire package of traits can be mathematically treated as one single trait.

Other connected models from genetic theory may be used as a starting point for further development of the theory underpinning linked cultural evolution. For example, models of the evolution of recombination rate (e.g. [33,34]) may provide a basis for more nuanced future models of the evolution of link formation and breakage ($a$ and $b$ in our model). Further, we have not considered here that cultural traits linked together may have synergistic effects—in other words, the benefit of the whole cultural package may be more than the sum of the benefits of the component parts. A deep theory of the interaction between genes and the synergistic effects that they may have, known as epistasis, exists and may provide a path forward for the study of cultural linkage.

## (c) Inference of underlying processes of cultural transmission

Many studies are interested in inferring the underlying processes of cultural transmission from the available data, often population-level frequencies of cultural variants. Recent studies have clearly shown that there exist theoretical limits to inferring processes of cultural transmission from population-level patterns: one should not expect a one-to-one mapping between population-level statistics and the underlying transmission process as different scenarios can lead to comparable population-level patterns [9,27,35,36]. Here, we add to the list of potentially confounding factors. We have shown that low diversity may not be indicative of a departure from unbiased transmission but of the existence of links. Consequently, if we analyse a cultural system where package transmission is possible with statistics that assume independent trait transmission, we may wrongly interpret low pairwise difference values as evidence against unbiased transmission or as evidence for pay-off-biased or conformist-biased transmission (figure 4b). Additionally, we have shown that through the process of hitchhiking, a neutral associated trait can possess the signature of a functional trait during a cultural sweep and for some time after the sweep (figures 4c and 5a–c).

These results make the identification of links and packages in real-world situations an important task. While there are established procedures to collect the type of sequence data needed to identify linkage in a genetic system, it is difficult to collect such detailed data in a cultural context.

## (d) Implications for cultural evolutionary models

There is a dearth of empirical studies aiming to identify the presence of links between cultural traits, but there is no reason to assume that links are rare, and evidence for hitchhiking does exist (e.g. [37]). We urgently need theory that contains, among other things, a psychological mechanism of link formation and maintenance, a description of the transmission of linked traits and the links between them, and mechanisms of how innovation interacts with linked traits and how novel cultural traits are incorporated in cultural packages. The development of such models should be the focus of immediate future work as they will further clarify when the existence of links appreciably alter evolutionary dynamics (e.g. [38,39]). However, the development of such theoretical models must go hand in hand with the development of statistical procedures that can identify the presence of links and the transmission of packages in real cultural systems, based on available data.

Such a complete theory of linked cultural transmission may prove crucial to our understanding of some basic metrics of cultural evolution in a variety of domains. As mentioned above, complex human technology represents an extreme of large, relatively stable cultural packages. However, the story of the evolution of technologies from chimpanzee nut cracking to complex human artefacts like axes and arrows is the story of the formation and maintenance of links between once-distinct cultural objects. Thus, to understand cultural complexity in its many forms, we must continue to develop a solid theory of cultural linkage.

Data accessibility. The codes are archived on the Dryad Digital Repository: https://doi.org/10.5061/dryad.zkh18935z [25].

Authors' contributions. D.J.Y. conceived of the study, built the model, ran the simulations, carried out the analysis and drafted the manuscript. L.F. and A.K. provided critical input to model design, analyses and figures and heavily edited the manuscript. All the authors gave

final approval for publication and agreed to be held accountable for the work performed therein.

Competing interests. We declare we have no competing interests.

Funding. We received no funding for this study.

Acknowledgements. We thank the editor, two anonymous reviewers, John Bunce, Adam Powell and members of the Department of Human Behavior, Ecology and Culture at the Max Planck Institute for Evolutionary Anthropology for constructive discussions and criticisms which helped improving this paper.

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
