## [Reviewer comments · Proceedings of the Royal Society B: Biological Sciences]

Review History

RSPB-2019-1951.R0 (Original submission)

Review form: Reviewer 1

Recommendation

Accept with minor revision (please list in comments)

Scientific importance: Is the manuscript an original and important contribution to its field?

Excellent

General interest: Is the paper of sufficient general interest?

Good

Quality of the paper: Is the overall quality of the paper suitable?

Excellent

Is the length of the paper justified?

Yes

Should the paper be seen by a specialist statistical reviewer?

Yes

Do you have any concerns about statistical analyses in this paper? If so, please specify them explicitly in your report.

No

It is a condition of publication that authors make their supporting data, code and materials available - either as supplementary material or hosted in an external repository. Please rate, if applicable, the supporting data on the following criteria.

Is it accessible?

N/A

Is it clear?

N/A

Is it adequate?

N/A

Do you have any ethical concerns with this paper?

No

Comments to the Author

The authors explore the implications of cultural trait linkages on the generation of cultural diversity and on the possibility of determining cultural evolutionary processes from cultural trait frequencies, using a simulation model. The implications of their work are extremely important for studying the dynamics of cultural evolution, as human “cultures” are clearly made up of highly interdependent linked cultural traits (such that treating cultural traits like beanbag genetics would be inappropriate). In other words this paper has high scientific significance. It is well written, and an important contribution to the literature.

A major finding is that low cultural diversity values may result not from biased transmission but the existence of links between cultural traits; also that the distribution and patterning of (fitness) neutral traits can bear the signature of a functional trait, as a consequence of their “hitchhiking” along with another trait. While I am not qualified to assess the technicalities of the simulation (ie the technical quality), the steps outlined, and the assumptions, to me seem reasonable (with one exception, see below) – albeit highly simplified (which is of course fine for a modeling/simulation approach). Their findings regarding hitchhiking and innovation make intuitive sense and are well substantiated with their simulation analysis.

I do however have three concerns with this as a Proc B paper that can, I am sure, be addressed.

First, I wonder whether it is realistic to assume that links between cultural traits are formed at random, since to me it seems highly likely that functionally interdependent traits (i.e., traits that only confer fitness if in association with each other) are likely cores to “trait packages”, see the Boyd et al 1995 paper mentioned below.

Second, the current framing of the paper does not seem very well suited to Proc B. For example, the authors claim that their major contribution is to provide insights into how linked traits can interfere with the signature of cultural evolutionary processes (and the patterning of cultural frequency data (:68)). I personally found it very interesting, but I am not sure this is a compelling read for biologists more generally, although I may be wrong – certainly important kinks in deciphering evolutionary processes are uncovered. My sense is that if the paper were framed more substantially around the patterning of cultural diversity (and the relevant literature, see my

third concern), its significance would be more broadly recognized. On rereading the paper before submitting this review it struck me that some of the material in the final section of the Discussion could be brought forward to perhaps make the paper more broadly appealing.

Finally, some very obvious linkages to pertinent literature seem to be missed; for example “The strong effects of linkage on genetic evolution hint at the possibility that links between cultural traits might change evolutionary dynamics in unexpected ways” (39-40). This question was addressed conceptually by Boyd et al (1997, albeit with a different set of inferences). This paper laid out a set of scenarios for how traits might be linked that would seem to anticipate some of the ideas in this paper. Also “one should not expect a one-to-one mapping between population-level statistics and the underlying transmission process as different scenarios can lead to comparable population-level patterns [11,19,20]” 439-440. Stephen Shennan (Shennan 2009, 2011) has a whole book on this, based on empirical work. Finally the complexity of bundles of cultural traits and their interrelations (cf lines 21-22) have been closely studied by Jordan (Jordan 2014). I feel the authors could give their paper more traction if it were more firmly embedded in these parts of the cultural evolutionary literature.

That said, this is a fine piece of work that should be given strong consideration by Proc B.
 Boyd R, Borgerhoff Mulder M, Durham WH, Richerson PJ. 1997. Are cultural phylogenies possible? Pages 355-386 in Weingart P, Mitchell SD, Richerson PJ, and Maasen S, editors. *Human by Nature: Between Biology and the Social Sciences*. Erlbaum, Mahwah, NJ.
 Jordan P 2014. *Technology as Human Social Tradition: Cultural Transmission among Hunter Gatherers*. University of California Press, Berkeley.
 Shennan S. 2009. Pattern and process in cultural evolution: An introduction. Pages 1-18 in Shennan S, editor. *Pattern and Process in Cultural Evolution*. University of California Press, Berkeley.
 Shennan S. 2011. Property and wealth inequality as cultural niche construction. *Philosophical Transactions of the Royal Society B* 366:918-926.

Review form: Reviewer 1 (L.S. Premo)

Recommendation

Major revision is needed (please make suggestions in comments)

Scientific importance: Is the manuscript an original and important contribution to its field?

Good

General interest: Is the paper of sufficient general interest?

Good

Quality of the paper: Is the overall quality of the paper suitable?

Good

Is the length of the paper justified?

Yes

Should the paper be seen by a specialist statistical reviewer?

No

Do you have any concerns about statistical analyses in this paper? If so, please specify them explicitly in your report.

No

It is a condition of publication that authors make their supporting data, code and materials available - either as supplementary material or hosted in an external repository. Please rate, if applicable, the supporting data on the following criteria.

Is it accessible?

Yes

Is it clear?

Yes

Is it adequate?

Yes

Do you have any ethical concerns with this paper?

No

Comments to the Author

I have had the opportunity to read and think about the interesting manuscript entitled, "Cultural linkage: the influence of package transmission on cultural dynamics." The authors' introduction highlights the importance of three questions – "how do links between cultural traits emerge? How are they transmitted? And how should the existence of links change our expectations for the outcome of cultural evolutionary processes?". They make it clear that they set out to address only the third question in this paper. They employ an agent-based model (the Matlab code is made available but there is no mention of the model description following something akin to the ODD protocol introduced by Grimm et al.) to run simulation experiments designed to inform us of how random linkage between traits might impact cultural evolutionary dynamics in a constant sized finite population of $N=1000$ individuals. They investigate both unbiased and payoff-biased oblique transmission of targeted traits (and those that are linked to the targeted traits). All traits not included in the interaction partner's cultural "package" are transmitted vertically from the previous timestep. They further assume that the links that bind traits into "packages" can also be transmitted from an interaction partner, though this transmission is imperfect and some links can be broken along the way. They allow for all traits to undergo "innovation" at each time step. They use mean pairwise difference to investigate the effects of the probability of links being broken (b) and the rate at which new links are regenerated (a) on cultural diversity. They investigate hitchhiking under neutral conditions and conditions in which selection against the detrimental associated trait is fairly weak compared to selection for the beneficial target trait. Finally, they assess to what extent equifinality clouds one's ability to recognize unbiased transmission from cultural diversity if one incorrectly assumes that all traits are independently transmitted (i.e., no links) when in fact they were linked to some extent.

Generally speaking, I like the direction of this study. I think the question the authors choose to address is a good one, and I wholeheartedly agree that we should spend more time and energy studying how cultural linkage will affect cultural evolutionary dynamics. I also like the analysis they have done to address equifinality and the difficulty of identifying individual-level process from population-level pattern. Having said that, my overall impression after reading the study is that the model they present includes more than is needed to address the question they set out to answer. In particular, I do not think the stated goal requires that links between traits are transmitted, broken, and regenerated through the course of a simulation run. For one, it is unclear under what conditions such links are transmitted empirically, at least in humans. But

more importantly, it seems the main question posed in this study can be addressed without complicating the model with the transmission of links. I found myself wondering how much of the results are a function of the rather complicated, asymmetrical, and arbitrary rules governing when and how links are broken during transmission relative to how much is a function of the simple presence and number of links (which seems to me to be the main aim of the study). This ambiguity clouds my understanding of how well the paper addresses the question of interest, which has only to do with the presence of links and not their formation and dynamic change through time. Don't get me wrong, how these cultural packages form and change through time as a function of transmission is an interesting question and one worth tackling. However, it is not the question they set out to address here. In my opinion, including link transmission complicates the task of addressing the question of interest.

I have included a number of line-by-line comments and suggestions below, and I hope they are of some use to the authors. I hope that most of the suggestions are on point, and I apologize if I simply misunderstood or missed something when reading the manuscript – although I have dedicated a good chunk of my time to this, it would always be nice to have more time to spend with reviews. My most substantive comment/suggestion, however, is that in my opinion the model would be improved by simplifying it even more than it currently is. The goal of this simplification is to address the stated research question more directly and clearly. In particular, I would recommend getting rid of the assumption that links are transmitted. This would allow the authors to jettison some of the parameters (b and a) associated with link breakage and creation. In their place the authors could simply use one parameter (l) to control for the number of (static) links found in each individual throughout the course of each simulation. I would also recommend setting the variable c to 1 instead of 0.99, which means they could go ahead and eliminate c as well.

I foresee at least two major upsides to removing the parameters a , b and c along with the assumption that links are dynamic. First, I think the results would be more directly applicable to the question at hand. Second, removing a and b would free them up to run simulations with different values of h and k (at the moment, they only investigate $h=5$ and $k=4$). It would be interesting to see if the magnitude of the effects of cultural linkage on cultural diversity vary with the total number of traits and the number of variants possible at each trait.

In sum, at the moment, the model seems more complicated than it needs to be to address the research question. I would recommend using a simpler model to address the stated question in this paper. For the purposes of a second and separate study, they could always modify the model to address how dynamic cultural packages affect cultural diversity if they so desire. It seems to me they have included aspects of the second model that are not needed to address the question they set out to address in this first paper. The model can be pared down to better fit the needs of this project and doing so will likely make the model more general and the results easier to understand in the context of the research question.

line-by-line comments/questions/suggestions

line 10: "that allows links to form and break between cultural traits."

As I discussed above and describe in more detail below, I don't think it is necessary to assume that links break and regenerate to address the central question of this paper. In fact, the study might be improved (made simpler and easier to interpret) if the assumption that links are transmitted is dropped entirely.

Line 15: There is no mention of the equifinality issue in the abstract even though I think that is

one of the more important points to come out of this paper. I'd recommend adding a sentence to the abstract that summarizes that part of the study.

line 18: "Many definitions of complexity rely on some quantification of a trait's component parts and the extent to which those parts are integrated with one another (e.g. [1])."

The authors might benefit from being more precise here. If my memory serves me well, Oswalt focuses on food-getting technology and writes about technounits (the "component parts" in the quote above), but I don't think he would have called a harpoon a "cultural trait." He might use the term "cultural trait" to refer to an individual technounit (though I don't recall him doing that), but, even then, he makes no distinction between different variants of a technounit. Whether the technounit is made of leather, string, or sinew, it would all be the same to Oswalt as long as the component (the technounit) played the same role in the function of the tool it was a part of. In short, even if one equates Oswalt's technounit with a discrete culturally transmitted component of a tool with a finite number of possible variants (string, sinew, leather) – which is what the authors model in their paper – Oswalt's definition of technounit makes no distinction between those qualitatively different variants. In short, his framework is blind to cultural variation in each technounit. For that reason, Oswalt's work may not be the best example to cite here.

line 21: "This definition suggests that strong links between once-independent components may be a fundamental feature of complex human technology."

Perhaps, or that may be reading too much into the passage. After all, an assemblage of practices does not necessarily require links between them (although I suppose that it often does, as the authors imply).

Lines 27-28. "And how should the existence of links change our expectations for the outcome of cultural evolutionary processes?"

Yes, good question. That is the focus of this study.

line 43: "Any cultural trait could be linked to any other producing clusters of traits tightly linked in a network and difficult to separate."

Perhaps I'm taking this too literally, but... Surely there are examples of traits that could not be linked because they are mutually incompatible, no? The details involved in preparing a beef steak dinner probably will not be linked with being vegan. Atheists are unlikely to know which religious ceremonies or prayers to recite on certain days or at certain events. My point is that it may be too general to state that any trait can be linked to any other; identity is complicated and some traits may not possibly be linked to others depending on the variant expressed at one or the other. Clearly this is not a major critique, but perhaps this sentence can be scaled back to allow for the possibility of non-random links.

line 50: "In the following, for each of these three essential components, we make the simplest possible assumptions to begin to understand the effect of links on cultural evolutionary dynamics."

I don't agree with the claim that "the simplest possible assumptions" are used here. In particular, it is my opinion that the assumption that links are transmitted is not only rather complicated but also unnecessary to address the main question of the study.

Page 2-3: "Our model assumes that links form at random at a fixed rate between any two traits in

an individual's cultural repertoire. This is consistent with a mechanism of link formation where, for example, a role model demonstrates actions in sequence."

The second sentence does not seem to follow from the first. The process described — a model demonstrating something in sequence (say knapping a stone tool) — does not seem like an illustration of how any 2 traits could become linked. In fact, it seems to illustrate a case where the linkage would be quite non-random (if I want to knap a stone tool, I will probably have to copy the traits that are linked in the knapping process exhibited by the model rather than one of the behaviors exhibited as well as the kinship term the role model uses to refer to his mother's brother. Of the other mechanisms listed at the end of this paragraph, perhaps only prestige-biased transmission could conceivably link totally unrelated traits. Even causal reasoning is likely to link traits non-randomly (contrary to Pavlov).

Page 3, lines 62-72. This is a very clear paragraph laying out the goals of the paper.

Page 4 "Each individual can form a link between any two traits, ..."
With what probability? When?

Page 4, lines 86 - 101. I take it an individual's "payoff" (Eq 2) is meant to be synonymous with its "fitness" or perhaps cultural influence and that $s_i > 0$ is meant to represent the strength of (cultural) selection acting on different variants of the trait, but I'm not 100% sure if that is what the authors mean. If so, it might be worth making that connection explicit with a sentence or two so as not to leave the reader guessing about the terminology. If not, then some text should be added that explains the distinction.

Page 5, line 107: I assume the payoff-based interaction does not allow an individual to pair with itself. Is that the case?

line 109: "Multiple individuals may choose the same interaction partner."

This makes sense, but it raises a related question that I do not see the answer to in the text: how/when do individuals update their variant value at each transmitted trait? Say Bob is chosen as the interaction partner by two others: Sally and Tate. Bob's interaction partner is Frieda, from whom he tries to copy trait 1. Further assume that Sally and Tate both attempt to copy Bob's variant at trait 1. However, Sally attempts to copy Bob before Bob copies trait 1 from Frieda, while Tate attempts to copy Bob after Bob's interaction with Frieda. Assuming the variant Bob acquired from Frieda was different from his previous variant at trait 1, Sally and Tate would have copied different variants from Bob if individuals update variant values in real time (i.e., asynchronously). Alternatively, if all individuals update synchronously, then Sally and Tate would have copied the same variant value from Bob, because Bob would not have updated his variant at trait 1 (the one he got from Frieda after teaching Sally but before teaching Tate) until the end of the time step. In my opinion, synchronous updating (i.e. everyone in generation t attempts to copy the variants displayed by their interaction partner in generation $t-1$) would make the results easier to interpret, but I can't tell what was done in the case of this model. Please note that I did not look at the code to find out, though of course the answer is there. Perhaps it is in the text and I just missed it somehow, but the reader should be get this info without looking into the code.

Page 5: "Each variant is copied successfully with probability c , while unsuccessful copying events mean that the focal individual keeps its original variant."

I admit that I find this decision a bit confusing. If I understand correctly, $1-c$ does not represent copying error, per se. That is, $1-c$ is not meant to be the cultural analog of mutation. Instead, c

simply represents the probability that a given trait (the chosen one or one linked to the chosen one) will get transmitted from the interaction partner to the focal individual. I'm not sure what its utility is here. More specifically, why would c ever be less than 1? And if c doesn't represent copying error, then, for better or for worse, copying error is missing from this model, which might be an oversight. If $c=0$, nothing gets transmitted obliquely and the model defaults to vertical transmission of unlinked traits. So, all in all, I take it that c represents the strength of oblique transmission relative to vertical transmission of the chosen trait and those that are linked to the target trait. If that is the case, perhaps it is best to explain it that way (or perhaps I misunderstood the text). To isolate the effects of linkage, I would simply set c to 1 for the purposes of this paper, which would mean that I would not include c at all. Removing c (or setting c equal to 1) would mean that all of the traits linked to the target trait would be passed via oblique transmission and all traits outside of this "package" would be passed via vertical transmission. Wouldn't this also make for a cleaner comparison to the W-F expectations under conditions in which oblique transmission is unbiased and all 5 traits are linked to each other? The value of c used in the paper is very close to 1 (0.99), but I do not recall an explanation for that value or for why it is not varied in the experimental design. I think I would simply get rid of c , which is the same as setting it to 1 instead of 0.99.

Page 5 "the focal individual acquires each link in the package with probability $1-b$, so that with probability b a link in the transmitted package is broken."

I continue to question the decision to make links transmittable. I question this decision not because the assumption is "unrealistic," but because it seems to make for a weaker experimental design. Of course, the package of variants needs to be transmitted obliquely – that much is crystal clear. But I do not think the links need to be transmitted to address the following question: "And how should the existence of links change our expectations for the outcome of cultural evolutionary processes? In this paper we focus on the last question and present a model where cultural traits can be linked together and transmitted as a package."

To my mind, there is no reason to make the links transmittable to address that question. Dropping that notion would also allow you to get rid of the parameter, b . It would also do away with the asymmetry in how the transmission of links is dealt with: at the moment b applies only to links that are present in the interaction partner and not to the links that the focal individual has but the partner does not – the latter are essentially overwritten by the links present in the partner. Ego's links that connect to traits outside of its interaction partner's linked constellation are also erased. This is why the number of links drops to 0 when $a=0$. If one does not need to worry about transmitting links (which is not really the stated purpose of this paper and actually seems to be a more difficult and involved question than what is being addressed here), then one does not need b at all. And if links cannot be broken because they are not transmitted, then there is no need for a . It seems to me the experimental design would be much simpler if you do not worry about the transmission of links. You could introduce a new parameter, l , to represent the number of links in each individual. Those links would be constant in each individual throughout each simulation run. You could vary l between 0 and 10 to arrive at the same conditions illustrated in Figure 2 without bothering with a and b at all. In short, it strikes me that while including the transmission of links introduces the need for a number of variables and tough decisions about how links can be passed and broken, it does not lead to insights great enough to outweigh the costs. For these reasons, I'd advise against modeling the links as transmittable. This results in a model with fewer moving parts, fewer parameters, and fewer assumptions but clearer results for addressing the question of interest. Save the transmission of links for a second step.

line 132: "After transmission, new links randomly form at the rate of association, a between any of an individual's traits that are not linked."

Again, I'd get rid of the notion that links are transmitted. It does not seem necessary to address the stated question. If links are not transmitted, they cannot be broken. If they cannot be broken, then there is no need to regenerate them through this process involving a . Each individual will have the same links throughout the simulation.

line 133: "Individuals may then innovate a variant for each trait with probability μ ."

So, this model represents "innovation" rather than copying error. I'm concerned that "innovation" might connote progress or guided variation, while the algorithm provided shows that any potential variant value is equally likely to be adopted as a result of "innovation" during this process. Is this the case? I think I would call this "copying error" so as not to imply that "innovations" are directed to better values (in fact, in the case of the neutral traits, none of the variant values are more fit than any other). Because some of the traits are obtained obliquely (those copied from the interaction partner) and some are obtained vertically (those not copied from the interaction partner but copied from ego's previous generation), the term copying error is still legit in this case, I think.

Why was only one innovation rate tested? Why not more? If you get rid of parameters a , b , and c , it might be interesting to test additional values of μ , h , and k in your experimental design. I would be very interested to see how the results vary with greater h and k , for instance.

lines 145-150: I see why those "adjustments" are needed to address the hitchhiking question, but they are so heavy-handed that they raise the question of why even conduct the burn-in period in the first place? It seems like the population can be set as needed in this case without the burn-in period.

lines 169-170: "In contrast, if π_i has a value close to 1 then the variants occur with almost equal frequency in the population."

I think it might be worth modifying this slightly to read something like "higher values represent populations in which all variants occur with equal frequency" because just how closely the value approaches 1 when all variants are represented equally is dependent upon the number of variants possible at the trait. For example, for the conditions of this model, if each of the 4 variants of a trait appeared in exactly 250 of the $N=1000$ individuals, the resulting value would be 0.751. 0.751 is obviously closer to 1 than 0 is, but one might not consider 0.751 "close to 1" in an absolute sense even though all 4 variants are represented equally in the population. Now imagine there were 100 possible variants and each was displayed by 10 of the $N=1000$ individuals. Now the value (0.991) is indeed close to one in the absolute sense even though the variants are just as evenly represented among the 100 variants as in the previous case with 4 variants. If only 2 possible variants are distributed equally, the value isn't very close to 1 in the absolute sense.

line 195: Is Figure 2 depicting the mean pairwise difference of just one trait or the mean of the mean pairwise differences of all 5 traits? This is not specified in the caption.

line 206: "When $n=0$ there is only innovation and no transmission, and so the pairwise difference is maximized."

This might be a semantic quibble, but isn't there still vertical transmission in this case instead of no transmission? $n=0$ simply means none of the traits were obtained via oblique transmission and all traits were obtained via vertical transmission from the previous timestep.

lines 210-212: “To do this, we scale the rate of innovation by h/nc to account for the fact that innovation occurs at a constant rate in each timestep while the traits are, on average, only successfully transmitted every h/nc timesteps.”

This seems like an example of where the analysis would be simpler if the authors eliminated the variable c altogether.

lines 227-240: This is an interesting paragraph. It makes sense that variable package size would also have an interesting effect on the transmission of some traits (those that are located within a large cluster) over others (those located in a smaller cluster). If the authors were to take my advice and keep the number of links constant throughout each simulation by dropping the notion that links are transmitted (and thus dropping the need for a and b), then within each simulation there would probably be less variance in package size – e.g., 4 links could connect either 4 traits or 5 traits – than what they found in their simulations so far. Simplifying their model in this way might reduce the magnitude of the effect of variance in trait cluster size they describe here. That would be a cost, but the benefits of simplifying the model might outweigh this cost.

lines 231-233: “In Fig. 3B we show the variance of the package size – where the variance is high the benefit of being in a large package is highest. The variance becomes lower when a is high or b is low because in these cases all traits are linked in large packages.”

But it looks to me that the variance can also be low (note: the y -axis of fig. 3b should be flipped so the values increase as they go up) when a is low. And the results show that variance can still be quite high when b is assigned the lowest value tested here. In short, the description provided in the text seems incomplete, if not contradicted, by the results presented in fig. 3b.

lines 238-240: “This suggests an important interplay between package size and package stability over time in driving cultural evolution with linked traits.”

Yes, this is an intuitive but interesting and important point worth highlighting.

lines 241-256: This is a very good point about equifinality. I wonder if the results would be similar in the simpler model I’ve suggested in which links are not transmitted and a , b , and c are jettisoned? If not, then that would be a bit concerning.

line 259: “conformist-biased transmission”

Where did talk of conformist transmission come from? This has not been discussed at all in the paper to this point. It seems a bit jarring to throw it in at this point.

line 265: “See Appendix, Section 2 for how conformity is modelled.”

I would argue in this case that one should not have to go to the appendix to see how conformity is modelled. If it is important enough to be included in one of the main figures in the text, it should probably be explained in the main text along with the other major components of the model.

line 284: “We have shown that in situations of low (but positive) or intermediate link frequency the level of cultural diversity as measured by the pairwise difference is lower compared to the Wright-Fisher expectation (cf. Figs. 2A,B).”

Have I misunderstood something, because Fig. 2B shows that this is not the case for $b=0.1$? When

$b=0.1$, the pairwise difference remains higher than the W-F expectation when a is low (but positive). As a related side note, it would be nice to see the error bars around the means presented in Fig. 2.

line 314: "We show only estimates of this mean time for parameter constellations where hitchhiking occurred in sufficiently many simulations (at least 30) and..."

This sounds vague and somewhat arbitrary. What makes 30 "sufficiently many," and out of how many?

line 375: "Further, hitchhiking becomes less likely the more detrimental the associated variant."

OK, but does it still happen to some degree as long as $s_1 > s_2$? Or does the fact that the links are broken frequently in the current model mean that hitchhiking won't occur at all if the associated trait is detrimental? This is another place where allowing links to be passed, broken, and created during the simulation may make things more complicated than they need to be this early in the game.

line 384: "We note that by design the modelling assumptions are as simple as they can be to begin to understand the effect of links on cultural evolution..."

I disagree with "as simple as they can be"; the model can be simplified further by getting rid of the idea that links are transmitted according to what turns out to be a rather complicated set of rules. I am of the opinion that the costs of including link transmission outweigh the benefit at this early stage.

line 386: "We assume that links form at random at a fixed rate between any two traits in an individual's cultural repertoire and can be broken through the process of cultural transmission."

My major suggestion for this study is to do away with those assumptions altogether; to start simpler with this first model on the topic. It might be worth returning to the issue of how dynamic link formation might influence cultural diversity, but that seems a separate and more difficult question than the stated focus of this paper.

line 388: "Whether this assumption is appropriate depends on the study system and the nature of specific links."

Fair enough, but I'd also argue that this assumption may not even be appropriate for the question at hand, regardless of any actual study system one might be interested in down the road.

line 397: "The effects of links on pairwise difference interact with the rate of innovation: the smaller the rate, the lower the pairwise difference for low or intermediate link frequencies (see Appendix, Section 1, Fig. A1.1A,B)"

Is this point important enough to include the figure that supports it in the main paper rather than in the appendix? I think it might be.

line 418: "In our model the linkage patterns can vary between individuals and are copied along with trait variants, allowing variants in large packages to spread faster than those in smaller ones."

As described above, I am not sold on the need to allow for links to be transmitted in the current version of the model. I think there would still be a way to address the issue of how a larger

number of links can enhance the speed with which a trait in a larger package can spread versus a trait in a smaller package by simply controlling for the number of links found in all individuals at the start of the simulation (e.g., all individuals in a run would have 3 links and they would keep those for the entire simulation regardless of their interaction partners' links) as an experimental parameter called "l" for links, but disallowing those links to be passed, broken, or regenerated through the course of a simulation. I may be wrong, but to my mind, that would be a simpler and cleaner way to address the central issue of this paper.

lines 438-442: These are some very interesting thoughts for future work.

Decision letter (RSPB-2019-1951.R0)

27-Sep-2019

Dear Dr Yeh:

Your manuscript has now been peer reviewed and the reviews have been assessed by an Associate Editor. The reviewers' comments (not including confidential comments to the Editor) and the comments from the Associate Editor are included at the end of this email for your reference. As you will see, the reviewers and the Editors have raised some concerns with your manuscript and we would like to invite you to revise your manuscript to address them.

Research ethics:

Use of animals and field studies:

Please submit a copy of your revised paper within three weeks. If we do not hear from you within this time your manuscript will be rejected. If you are unable to meet this deadline please let us know as soon as possible, as we may be able to grant a short extension.

Best wishes,
Dr Sasha Dall
<mailto:proceedingsb@royalsociety.org>

Associate Editor

Board Member: 1

Comments to Author:

Douhan et al.'s "Cultural linkage: the influence of package transmission on cultural dynamics" is a well written model exploring the role of links between cultural traits in the dynamics of cultural evolution. It certainly fills a hole in a literature that has wide-reaching consequences.

I was positively inclined towards the MS, and so were both reviewers. However, one reviewer asks for a major revision. This reviewer had many other points, which I will not repeat here, but makes a convincing case that the main question asked can be better/clearer answered by a reduced model - namely one in which the trait linkages themselves are not culturally transmitted. I agree with this reviewer, and I therefore think that the authors should aim to include such a model in their MS (though I think the MS still should contain their current model, linked to the slightly different question of what happens when the trait linkages are themselves transmitted). The other reviewer points out (rightly) that some additional, relevant literature need to be acknowledged and discussed. In addition, the other reviewer points out that trait linkages may be non-random (and again, I agree).

Best regards

Claudio Tennie

Reviewer(s)' Comments to Author:

Referee: 1

Comments to the Author(s)

The authors explore the implications of cultural trait linkages on the generation of cultural diversity and on the possibility of determining cultural evolutionary processes from cultural trait frequencies, using a simulation model. The implications of their work are extremely important for studying the dynamics of cultural evolution, as human "cultures" are clearly made up of highly interdependent linked cultural traits (such that treating cultural traits like beanbag genetics would be inappropriate). In other words this paper has high scientific significance. It is well written, and an important contribution to the literature.

A major finding is that low cultural diversity values may result not from biased transmission but the existence of links between cultural traits; also that the distribution and patterning of (fitness) neutral traits can bear the signature of a functional trait, as a consequence of their "hitchhiking" along with another trait. While I am not qualified to assess the technicalities of the simulation (ie the technical quality), the steps outlined, and the assumptions, to me seem reasonable (with one exception, see below) - albeit highly simplified (which is of course fine for a modeling/simulation approach). Their findings regarding hitchhiking and innovation make intuitive sense and are well substantiated with their simulation analysis.

I do however have three concerns with this as a Proc B paper that can, I am sure, be addressed.

First, I wonder whether it is realistic to assume that links between cultural traits are formed at random, since to me it seems highly likely that functionally interdependent traits (i.e., traits that only confer fitness if in association with each other) are likely cores to "trait packages", see the Boyd et al 1995 paper mentioned below.

Second, the current framing of the paper does not seem very well suited to Proc B. For example, the authors claim that their major contribution is to provide insights into how linked traits can interfere with the signature of cultural evolutionary processes (and the patterning of cultural frequency data (:68)). I personally found it very interesting, but I am not sure this is a compelling read for biologists more generally, although I may be wrong – certainly important kinks in deciphering evolutionary processes are uncovered. My sense is that if the paper were framed more substantially around the patterning of cultural diversity (and the relevant literature, see my third concern), its significance would be more broadly recognized. On rereading the paper before submitting this review it struck me that some of the material in the final section of the Discussion could be brought forward to perhaps make the paper more broadly appealing.

Finally, some very obvious linkages to pertinent literature seem to be missed; for example “The strong effects of linkage on genetic evolution hint at the possibility that links between cultural traits might change evolutionary dynamics in unexpected ways” (39-40). This question was addressed conceptually by Boyd et al (1997, albeit with a different set of inferences). This paper laid out a set of scenarios for how traits might be linked that would seem to anticipate some of the ideas in this paper. Also “one should not expect a one-to-one mapping between population-level statistics and the underlying transmission process as different scenarios can lead to comparable population-level patterns [11,19,20]” 439-440. Stephen Shennan (Shennan 2009, 2011) has a whole book on this, based on empirical work. Finally the complexity of bundles of cultural traits and their interrelations (cf lines 21-22) have been closely studied by Jordan (Jordan 2014). I feel the authors could give their paper more traction if it were more firmly embedded in these parts of the cultural evolutionary literature.

That said, this is a fine piece of work that should be given strong consideration by Proc B.

Boyd R, Borgerhoff Mulder M, Durham WH, Richerson PJ. 1997. Are cultural phylogenies possible? Pages 355-386 in Weingart P, Mitchell SD, Richerson PJ, and Maasen S, editors. *Human by Nature: Between Biology and the Social Sciences*. Erlbaum, Mahwah, NJ.

Jordan P 2014. *Technology as Human Social Tradition: Cultural Transmission among Hunter Gatherers*. University of California Press, Berkeley.

Shennan S. 2009. Pattern and process in cultural evolution: An introduction. Pages 1-18 in Shennan S, editor. *Pattern and Process in Cultural Evolution*. University of California Press, Berkeley.

Shennan S. 2011. Property and wealth inequality as cultural niche construction. *Philosophical Transactions of the Royal Society B* 366:918-926.

Referee: 2

Comments to the Author(s)

I have had the opportunity to read and think about the interesting manuscript entitled, “Cultural linkage: the influence of package transmission on cultural dynamics.” The authors’ introduction highlights the importance of three questions – “how do links between cultural traits emerge? How are they transmitted? And how should the existence of links change our expectations for the outcome of cultural evolutionary processes?”. They make it clear that they set out to address only the third question in this paper. They employ an agent-based model (the Matlab code is made available but there is no mention of the model description following something akin to the ODD protocol introduced by Grimm et al.) to run simulation experiments designed to inform us of how random linkage between traits might impact cultural evolutionary dynamics in a constant sized finite population of N=1000 individuals. They investigate both unbiased and payoff-biased

oblique transmission of targeted traits (and those that are linked to the targeted traits). All traits not included in the interaction partner's cultural "package" are transmitted vertically from the previous timestep. They further assume that the links that bind traits into "packages" can also be transmitted from an interaction partner, though this transmission is imperfect and some links can be broken along the way. They allow for all traits to undergo "innovation" at each time step. They use mean pairwise difference to investigate the effects of the probability of links being broken (b) and the rate at which new links are regenerated (a) on cultural diversity. They investigate hitchhiking under neutral conditions and conditions in which selection against the detrimental associated trait is fairly weak compared to selection for the beneficial target trait. Finally, they assess to what extent equifinality clouds one's ability to recognize unbiased transmission from cultural diversity if one incorrectly assumes that all traits are independently transmitted (i.e., no links) when in fact they were linked to some extent.

Generally speaking, I like the direction of this study. I think the question the authors choose to address is a good one, and I wholeheartedly agree that we should spend more time and energy studying how cultural linkage will affect cultural evolutionary dynamics. I also like the analysis they have done to address equifinality and the difficulty of identifying individual-level process from population-level pattern. Having said that, my overall impression after reading the study is that the model they present includes more than is needed to address the question they set out to answer. In particular, I do not think the stated goal requires that links between traits are transmitted, broken, and regenerated through the course of a simulation run. For one, it is unclear under what conditions such links are transmitted empirically, at least in humans. But more importantly, it seems the main question posed in this study can be addressed without complicating the model with the transmission of links. I found myself wondering how much of the results are a function of the rather complicated, asymmetrical, and arbitrary rules governing when and how links are broken during transmission relative to how much is a function of the simple presence and number of links (which seems to me to be the main aim of the study). This ambiguity clouds my understanding of how well the paper addresses the question of interest, which has only to do with the presence of links and not their formation and dynamic change through time. Don't get me wrong, how these cultural packages form and change through time as a function of transmission is an interesting question and one worth tackling. However, it is not the question they set out to address here. In my opinion, including link transmission complicates the task of addressing the question of interest.

I have included a number of line-by-line comments and suggestions below, and I hope they are of some use to the authors. I hope that most of the suggestions are on point, and I apologize if I simply misunderstood or missed something when reading the manuscript – although I have dedicated a good chunk of my time to this, it would always be nice to have more time to spend with reviews. My most substantive comment/suggestion, however, is that in my opinion the model would be improved by simplifying it even more than it currently is. The goal of this simplification is to address the stated research question more directly and clearly. In particular, I would recommend getting rid of the assumption that links are transmitted. This would allow the authors to jettison some of the parameters (b and a) associated with link breakage and creation. In their place the authors could simply use one parameter (l) to control for the number of (static) links found in each individual throughout the course of each simulation. I would also recommend setting the variable c to 1 instead of 0.99, which means they could go ahead and eliminate c as well.

I foresee at least two major upsides to removing the parameters a, b and c along with the assumption that links are dynamic. First, I think the results would be more directly applicable to the question at hand. Second, removing a and b would free them up to run simulations with different values of h and k (at the moment, they only investigate h=5 and k=4). It would be

interesting to see if the magnitude of the effects of cultural linkage on cultural diversity vary with the total number of traits and the number of variants possible at each trait.

In sum, at the moment, the model seems more complicated than it needs to be to address the research question. I would recommend using a simpler model to address the stated question in this paper. For the purposes of a second and separate study, they could always modify the model to address how dynamic cultural packages affect cultural diversity if they so desire. It seems to me they have included aspects of the second model that are not needed to address the question they set out to address in this first paper. The model can be pared down to better fit the needs of this project and doing so will likely make the model more general and the results easier to understand in the context of the research question.

line-by-line comments/questions/suggestions

line 10: "that allows links to form and break between cultural traits."

As I discussed above and describe in more detail below, I don't think it is necessary to assume that links break and regenerate to address the central question of this paper. In fact, the study might be improved (made simpler and easier to interpret) if the assumption that links are transmitted is dropped entirely.

Line 15: There is no mention of the equifinality issue in the abstract even though I think that is one of the more important points to come out of this paper. I'd recommend adding a sentence to the abstract that summarizes that part of the study.

line 18: "Many definitions of complexity rely on some quantification of a trait's component parts and the extent to which those parts are integrated with one another (e.g. [1])."

The authors might benefit from being more precise here. If my memory serves me well, Oswalt focuses on food-getting technology and writes about technounits (the "component parts" in the quote above), but I don't think he would have called a harpoon a "cultural trait." He might use the term "cultural trait" to refer to an individual technounit (though I don't recall him doing that), but, even then, he makes no distinction between different variants of a technounit. Whether the technounit is made of leather, string, or sinew, it would all be the same to Oswalt as long as the component (the technounit) played the same role in the function of the tool it was a part of. In short, even if one equates Oswalt's technounit with a discrete culturally transmitted component of a tool with a finite number of possible variants (string, sinew, leather) – which is what the authors model in their paper – Oswalt's definition of technounit makes no distinction between those qualitatively different variants. In short, his framework is blind to cultural variation in each technounit. For that reason, Oswalt's work may not be the best example to cite here.

line 21: "This definition suggests that strong links between once-independent components may be a fundamental feature of complex human technology."

Perhaps, or that may be reading too much into the passage. After all, an assemblage of practices does not necessarily require links between them (although I suppose that it often does, as the authors imply).

Lines 27-28. "And how should the existence of links change our expectations for the outcome of cultural evolutionary processes?"

Yes, good question. That is the focus of this study.

line 43: “Any cultural trait could be linked to any other producing clusters of traits tightly linked in a network and difficult to separate.”

Perhaps I’m taking this too literally, but... Surely there are examples of traits that could not be linked because they are mutually incompatible, no? The details involved in preparing a beef steak dinner probably will not be linked with being vegan. Atheists are unlikely to know which religious ceremonies or prayers to recite on certain days or at certain events. My point is that it may be too general to state that any trait can be linked to any other; identity is complicated and some traits may not possibly be linked to others depending on the variant expressed at one or the other. Clearly this is not a major critique, but perhaps this sentence can be scaled back to allow for the possibility of non-random links.

line 50: “In the following, for each of these three essential components, we make the simplest possible assumptions to begin to understand the effect of links on cultural evolutionary dynamics.”

I don’t agree with the claim that “the simplest possible assumptions” are used here. In particular, it is my opinion that the assumption that links are transmitted is not only rather complicated but also unnecessary to address the main question of the study.

Page 2-3: “Our model assumes that links form at random at a fixed rate between any two traits in an individual’s cultural repertoire. This is consistent with a mechanism of link formation where, for example, a role model demonstrates actions in sequence.”

The second sentence does not seem to follow from the first. The process described – a model demonstrating something in sequence (say knapping a stone tool) – does not seem like an illustration of how any 2 traits could become linked. In fact, it seems to illustrate a case where the linkage would be quite non-random (if I want to knap a stone tool, I will probably have to copy the traits that are linked in the knapping process exhibited by the model rather than one of the behaviors exhibited as well as the kinship term the role model uses to refer to his mother’s brother. Of the other mechanisms listed at the end of this paragraph, perhaps only prestige-biased transmission could conceivably link totally unrelated traits. Even causal reasoning is likely to link traits non-randomly (contrary to Pavlov).

Page 3, lines 62-72. This is a very clear paragraph laying out the goals of the paper.

Page 4 “Each individual can form a link between any two traits, ...”
With what probability? When?

Page 4, lines 86 – 101. I take it an individual’s “payoff” (Eq 2) is meant to be synonymous with its “fitness” or perhaps cultural influence and that $s_i > 0$ is meant to represent the strength of (cultural) selection acting on different variants of the trait, but I’m not 100% sure if that is what the authors mean. If so, it might be worth making that connection explicit with a sentence or two so as not to leave the reader guessing about the terminology. If not, then some text should be added that explains the distinction.

Page 5, line 107: I assume the payoff-based interaction does not allow an individual to pair with itself. Is that the case?

line 109: “Multiple individuals may choose the same interaction partner.”

This makes sense, but it raises a related question that I do not see the answer to in the text: how/when do individuals update their variant value at each transmitted trait? Say Bob is chosen

as the interaction partner by two others: Sally and Tate. Bob's interaction partner is Frieda, from whom he tries to copy trait 1. Further assume that Sally and Tate both attempt to copy Bob's variant at trait 1. However, Sally attempts to copy Bob before Bob copies trait 1 from Frieda, while Tate attempts to copy Bob after Bob's interaction with Frieda. Assuming the variant Bob acquired from Frieda was different from his previous variant at trait 1, Sally and Tate would have copied different variants from Bob if individuals update variant values in real time (i.e., asynchronously). Alternatively, if all individuals update synchronously, then Sally and Tate would have copied the same variant value from Bob, because Bob would not have updated his variant at trait 1 (the one he got from Frieda after teaching Sally but before teaching Tate) until the end of the time step. In my opinion, synchronous updating (i.e. everyone in generation t attempts to copy the variants displayed by their interaction partner in generation $t-1$) would make the results easier to interpret, but I can't tell what was done in the case of this model. Please note that I did not look at the code to find out, though of course the answer is there. Perhaps it is in the text and I just missed it somehow, but the reader should be get this info without looking into the code.

Page 5: "Each variant is copied successfully with probability c , while unsuccessful copying events mean that the focal individual keeps its original variant."

I admit that I find this decision a bit confusing. If I understand correctly, $1-c$ does not represent copying error, per se. That is, $1-c$ is not meant to be the cultural analog of mutation. Instead, c simply represents the probability that a given trait (the chosen one or one linked to the chosen one) will get transmitted from the interaction partner to the focal individual. I'm not sure what its utility is here. More specifically, why would c ever be less than 1? And if c doesn't represent copying error, then, for better or for worse, copying error is missing from this model, which might be an oversight. If $c=0$, nothing gets transmitted obliquely and the model defaults to vertical transmission of unlinked traits. So, all in all, I take it that c represents the strength of oblique transmission relative to vertical transmission of the chosen trait and those that are linked to the target trait. If that is the case, perhaps it is best to explain it that way (or perhaps I misunderstood the text). To isolate the effects of linkage, I would simply set c to 1 for the purposes of this paper, which would mean that I would not include c at all. Removing c (or setting c equal to 1) would mean that all of the traits linked to the target trait would be passed via oblique transmission and all traits outside of this "package" would be passed via vertical transmission. Wouldn't this also make for a cleaner comparison to the W-F expectations under conditions in which oblique transmission is unbiased and all 5 traits are linked to each other? The value of c used in the paper is very close to 1 (0.99), but I do not recall an explanation for that value or for why it is not varied in the experimental design. I think I would simply get rid of c , which is the same as setting it to 1 instead of 0.99.

Page 5 "the focal individual acquires each link in the package with probability $1-b$, so that with probability b a link in the transmitted package is broken."

I continue to question the decision to make links transmittable. I question this decision not because the assumption is "unrealistic," but because it seems to make for a weaker experimental design. Of course, the package of variants needs to be transmitted obliquely — that much is crystal clear. But I do not think the links need to be transmitted to address the following question: "And how should the existence of links change our expectations for the outcome of cultural evolutionary processes? In this paper we focus on the last question and present a model where cultural traits can be linked together and transmitted as a package."

To my mind, there is no reason to make the links transmittable to address that question. Dropping that notion would also allow you to get rid of the parameter, b . It would also do away with the asymmetry in how the transmission of links is dealt with: at the moment b applies only

to links that are present in the interaction partner and not to the links that the focal individual has but the partner does not – the latter are essentially overwritten by the links present in the partner. Ego's links that connect to traits outside of its interaction partner's linked constellation are also erased. This is why the number of links drops to 0 when $a=0$. If one does not need to worry about transmitting links (which is not really the stated purpose of this paper and actually seems to be a more difficult and involved question than what is being addressed here), then one does not need b at all. And if links cannot be broken because they are not transmitted, then there is no need for a . It seems to me the experimental design would be much simpler if you do not worry about the transmission of links. You could introduce a new parameter, l , to represent the number of links in each individual. Those links would be constant in each individual throughout each simulation run. You could vary l between 0 and 10 to arrive at the same conditions illustrated in Figure 2 without bothering with a and b at all. In short, it strikes me that while including the transmission of links introduces the need for a number of variables and tough decisions about how links can be passed and broken, it does not lead to insights great enough to outweigh the costs. For these reasons, I'd advise against modeling the links as transmittable. This results in a model with fewer moving parts, fewer parameters, and fewer assumptions but clearer results for addressing the question of interest. Save the transmission of links for a second step.

line 132: "After transmission, new links randomly form at the rate of association, a between any of an individual's traits that are not linked."

Again, I'd get rid of the notion that links are transmitted. It does not seem necessary to address the stated question. If links are not transmitted, they cannot be broken. If they cannot be broken, then there is no need to regenerate them through this process involving a . Each individual will have the same links throughout the simulation.

line 133: "Individuals may then innovate a variant for each trait with probability μ ."

So, this model represents "innovation" rather than copying error. I'm concerned that "innovation" might connote progress or guided variation, while the algorithm provided shows that any potential variant value is equally likely to be adopted as a result of "innovation" during this process. Is this the case? I think I would call this "copying error" so as not to imply that "innovations" are directed to better values (in fact, in the case of the neutral traits, none of the variant values are more fit than any other). Because some of the traits are obtained obliquely (those copied from the interaction partner) and some are obtained vertically (those not copied from the interaction partner but copied from ego's previous generation), the term copying error is still legit in this case, I think.

Why was only one innovation rate tested? Why not more? If you get rid of parameters a , b , and c , it might be interesting to test additional values of μ , h , and k in your experimental design. I would be very interested to see how the results vary with greater h and k , for instance.

lines 145-150: I see why those "adjustments" are needed to address the hitchhiking question, but they are so heavy-handed that they raise the question of why even conduct the burn-in period in the first place? It seems like the population can be set as needed in this case without the burn-in period.

lines 169-170: "In contrast, if π_i has a value close to 1 then the variants occur with almost equal frequency in the population."

I think it might be worth modifying this slightly to read something like "higher values represent populations in which all variants occur with equal frequency" because just how closely the value approaches 1 when all variants are represented equally is dependent upon the number of

variants possible at the trait. For example, for the conditions of this model, if each of the 4 variants of a trait appeared in exactly 250 of the $N=1000$ individuals, the resulting value would be 0.751. 0.751 is obviously closer to 1 than 0 is, but one might not consider 0.751 “close to 1” in an absolute sense even though all 4 variants are represented equally in the population. Now imagine there were 100 possible variants and each was displayed by 10 of the $N=1000$ individuals. Now the value (0.991) is indeed close to one in the absolute sense even though the variants are just as evenly represented among the 100 variants as in the previous case with 4 variants. If only 2 possible variants are distributed equally, the value isn’t very close to 1 in the absolute sense.

line 195: Is Figure 2 depicting the mean pairwise difference of just one trait or the mean of the mean pairwise differences of all 5 traits? This is not specified in the caption.

line 206: “When $n=0$ there is only innovation and no transmission, and so the pairwise difference is maximized.”

This might be a semantic quibble, but isn’t there still vertical transmission in this case instead of no transmission? $n=0$ simply means none of the traits were obtained via oblique transmission and all traits were obtained via vertical transmission from the previous timestep.

lines 210-212: “To do this, we scale the rate of innovation by h/nc to account for the fact that innovation occurs at a constant rate in each timestep while the traits are, on average, only successfully transmitted every h/nc timesteps.”

This seems like an example of where the analysis would be simpler if the authors eliminated the variable c altogether.

lines 227-240: This is an interesting paragraph. It makes sense that variable package size would also have an interesting effect on the transmission of some traits (those that are located within a large cluster) over others (those located in a smaller cluster). If the authors were to take my advice and keep the number of links constant throughout each simulation by dropping the notion that links are transmitted (and thus dropping the need for a and b), then within each simulation there would probably be less variance in package size – e.g., 4 links could connect either 4 traits or 5 traits – than what they found in their simulations so far. Simplifying their model in this way might reduce the magnitude of the effect of variance in trait cluster size they describe here. That would be a cost, but the benefits of simplifying the model might outweigh this cost.

lines 231-233: “In Fig. 3B we show the variance of the package size – where the variance is high the benefit of being in a large package is highest. The variance becomes lower when a is high or b is low because in these cases all traits are linked in large packages.”

But it looks to me that the variance can also be low (note: the y-axis of fig. 3b should be flipped so the values increase as they go up) when a is low. And the results show that variance can still be quite high when b is assigned the lowest value tested here. In short, the description provided in the text seems incomplete, if not contradicted, by the results presented in fig. 3b.

lines 238-240: “This suggests an important interplay between package size and package stability over time in driving cultural evolution with linked traits.”

Yes, this is an intuitive but interesting and important point worth highlighting.

lines 241-256: This is a very good point about equifinality. I wonder if the results would be

similar in the simpler model I've suggested in which links are not transmitted and a, b, and c are jettisoned? If not, then that would be a bit concerning.

line 259: "conformist-biased transmission"

Where did talk of conformist transmission come from? This has not been discussed at all in the paper to this point. It seems a bit jarring to throw it in at this point.

line 265: "See Appendix, Section 2 for how conformity is modelled."

I would argue in this case that one should not have to go to the appendix to see how conformity is modelled. If it is important enough to be included in one of the main figures in the text, it should probably be explained in the main text along with the other major components of the model.

line 284: "We have shown that in situations of low (but positive) or intermediate link frequency the level of cultural diversity as measured by the pairwise difference is lower compared to the Wright-Fisher expectation (cf. Figs. 2A,B)."

Have I misunderstood something, because Fig. 2B shows that this is not the case for $b=0.1$? When $b=0.1$, the pairwise difference remains higher than the W-F expectation when a is low (but positive). As a related side note, it would be nice to see the error bars around the means presented in Fig. 2.

line 314: "We show only estimates of this mean time for parameter constellations where hitchhiking occurred in sufficiently many simulations (at least 30) and..."

This sounds vague and somewhat arbitrary. What makes 30 "sufficiently many," and out of how many?

line 375: "Further, hitchhiking becomes less likely the more detrimental the associated variant."

OK, but does it still happen to some degree as long as $s_1 > s_2$? Or does the fact that the links are broken frequently in the current model mean that hitchhiking won't occur at all if the associated trait is detrimental? This is another place where allowing links to be passed, broken, and created during the simulation may make things more complicated than they need to be this early in the game.

line 384: "We note that by design the modelling assumptions are as simple as they can be to begin to understand the effect of links on cultural evolution..."

I disagree with "as simple as they can be"; the model can be simplified further by getting rid of the idea that links are transmitted according to what turns out to be a rather complicated set of rules. I am of the opinion that the costs of including link transmission outweigh the benefit at this early stage.

line 386: "We assume that links form at random at a fixed rate between any two traits in an individual's cultural repertoire and can be broken through the process of cultural transmission."

My major suggestion for this study is to do away with those assumptions altogether; to start simpler with this first model on the topic. It might be worth returning to the issue of how dynamic link formation might influence cultural diversity, but that seems a separate and more difficult question than the stated focus of this paper.

line 388: "Whether this assumption is appropriate depends on the study system and the nature of specific links."

Fair enough, but I'd also argue that this assumption may not even be appropriate for the question at hand, regardless of any actual study system one might be interested in down the road.

line 397: "The effects of links on pairwise difference interact with the rate of innovation: the smaller the rate, the lower the pairwise difference for low or intermediate link frequencies (see Appendix, Section1, Fig. A1.1A,B)"

Is this point important enough to include the figure that supports it in the main paper rather than in the appendix? I think it might be.

line 418: "In our model the linkage patterns can vary between individuals and are copied along with trait variants, allowing variants in large packages to spread faster than those in smaller ones."

As described above, I am not sold on the need to allow for links to be transmitted in the current version of the model. I think there would still be a way to address the issue of how a larger number of links can enhance the speed with which a trait in a larger package can spread versus a trait in a smaller package by simply controlling for the number of links found in all individuals at the start of the simulation (e.g., all individuals in a run would have 3 links and they would keep those for the entire simulation regardless of their interaction partners' links) as an experimental parameter called "l" for links, but disallowing those links to be passed, broken, or regenerated through the course of a simulation. I may be wrong, but to my mind, that would be a simpler and cleaner way to address the central issue of this paper.

lines 438-442: These are some very interesting thoughts for future work.

Author's Response to Decision Letter for (RSPB-2019-1951.R0)

See Appendix A.

Decision letter (RSPB-2019-1951.R1)

11-Nov-2019

Dear Dr Yeh

I am pleased to inform you that your manuscript entitled "Cultural linkage: the influence of package transmission on cultural dynamics" has been accepted for publication in Proceedings B.

You can expect to receive a proof of your article from our Production office in due course, please check your spam filter if you do not receive it. PLEASE NOTE: you will be given the exact page

length of your paper which may be different from the estimation from Editorial and you may be asked to reduce your paper if it goes over the 10 page limit.

Open Access

Paper charges

Sincerely,

Dr Sasha Dall

Associate Editor:

Board Member

Comments to Author:

Thank you for your work on the manuscript (especially for the new appendix) and for the detailed responses to the reviewers' comments.

Appendix A

Associate Editor

Board Member: 1

Comments to Author:

Douhan et al.'s "Cultural linkage: the influence of package transmission on cultural dynamics" is a well written model exploring the role of links between cultural traits in the dynamics of cultural evolution. It certainly fills a hole in a literature that has wide-reaching consequences.

I was positively inclined towards the MS, and so were both reviewers. However, one reviewer asks for a major revision. This reviewer had many other points, which I will not repeat here, but makes a convincing case that the main question asked can be better/clearer answered by a reduced model - namely one in which the trait linkages themselves are not culturally transmitted. I agree with this reviewer, and I therefore think that the authors should aim to include such a model in their MS (though I think the MS still should contain their current model, linked to the slightly different question of what happens when the trait linkages are themselves transmitted). The other reviewer points out (rightly) that some additional, relevant literature need to be acknowledged and discussed. In addition, the other reviewer points out that trait linkages may be non-random (and again, I agree).

Best regards

Claudio Tennie

First, we would like to thank both reviewers and the associate editor for such thoughtful and thorough reviews, which we have no doubt will substantially improve the paper, both presentation and results. We are pleased that the reviewers see the value in the work and in our results. It seemed to us that the major point requiring clarification was the necessity for, and results of, the assumption of transmitted links. We have run some further simulations to address this along the lines suggested by reviewer 2 and also clarified why we think ‘transmitted links’ is a more useful and defensible assumption. These simulations are detailed in Appendix 1 and we address all other comments below.

Reviewer(s)' Comments to Author:

Referee: 1

Comments to the Author(s)

The authors explore the implications of cultural trait linkages on the generation of cultural diversity and on the possibility of determining cultural evolutionary processes from cultural trait frequencies, using a simulation model. The implications of their work are extremely important for studying the dynamics of cultural evolution, as human “cultures” are clearly made up of highly interdependent linked cultural traits (such that treating cultural traits like beanbag genetics would be inappropriate). In other words this paper has high scientific significance. It is well written, and an important contribution to the literature.

A major finding is that low cultural diversity values may result not from biased transmission but the existence of links between cultural traits; also that the distribution and patterning of (fitness) neutral traits can bear the signature of a functional trait, as a consequence of their “hitchhiking” along with another trait. While I am not qualified to assess the technicalities of the simulation (ie the technical quality), the steps outlined, and the assumptions, to me seem reasonable (with one exception, see below) – albeit highly simplified (which is of course fine for a modeling/simulation approach). Their findings regarding hitchhiking and innovation make intuitive sense and are well substantiated with their simulation analysis.

I do however have three concerns with this as a Proc B paper that can, I am sure, be addressed.

First, I wonder whether it is realistic to assume that links between cultural traits are formed at random, since to me it seems highly likely that functionally interdependent traits (i.e., traits that only confer fitness if in association with each other) are likely cores to “trait packages”, see the Boyd et al 1995 paper mentioned below.

This is an excellent point and the assumption that links form at random is an oversimplification and many more realistic ‘theories of link formation’ are possible. Indeed, these will be necessary to progress on this question. We tried, in the manuscript, to make this point clear (e.g. in lines 56-59, 372-373, 375-376). We chose the simplest possible assumption about link formation because little is known about how links between traits really form and how domain specific, for example, such link formation theories might be. To investigate the dynamics of linked traits we suggest that this is an adequate starting point.

However, the comment suggests that the justification for this decision needs to be made clearer in the manuscript and for that reason we have added further explanation and expanded explanations that were present. These are on lines 42, 63-64 and 374.

Second, the current framing of the paper does not seem very well suited to Proc B. For example, the authors claim that their major contribution is to provide insights into how linked traits can interfere with the signature of cultural evolutionary processes (and the patterning of cultural frequency data (:68)). I personally found it very interesting, but I am not sure this is a compelling read for biologists more generally, although I may be wrong – certainly important kinks in deciphering evolutionary processes are uncovered. My sense is that if the paper were framed more substantially around the patterning of cultural diversity (and the relevant literature, see my third concern), its significance would be more broadly recognized. On rereading the paper before submitting this review it struck me that some of the material in the final section of the Discussion could be brought forward to perhaps make the paper more broadly appealing.

Thank you. As with many interdisciplinary studies it can be difficult to decide which journal is its natural home. We settled on Proc B hoping to reach a diverse audience of those interested in cultural evolution including archaeologists and others. We have added the suggested

references, and moved some relevant material from the discussion to the introduction (lines 52-56) which we hope clarifies the motivation for the model.

Finally, some very obvious linkages to pertinent literature seem to be missed; for example “The strong effects of linkage on genetic evolution hint at the possibility that links between cultural traits might change evolutionary dynamics in unexpected ways” (39-40). This question was addressed conceptually by Boyd et al (1997, albeit with a different set of inferences). This paper laid out a set of scenarios for how traits might be linked that would seem to anticipate some of the ideas in this paper. Also “one should not expect a one-to-one mapping between population-level statistics and the underlying transmission process as different scenarios can lead to comparable population-level patterns [11,19,20]” 439-440. Stephen Shennan (Shennan 2009, 2011) has a whole book on this, based on empirical work. Finally the complexity of bundles of cultural traits and their interrelations (cf lines 21-22) have been closely studied by Jordan (Jordan 2014). I feel the authors could give their paper more traction if it were more firmly embedded in these parts of the cultural evolutionary literature.

Thank you for pointing us to these papers and chapters. They approach the subject from a slightly different but certainly relevant viewpoint. We now include relevant points from the work of these authors on lines 45-50 and 416.

That said, this is a fine piece of work that should be given strong consideration by Proc B.

Pages 355-386 in Weingart P, Mitchell SD, Richerson PJ, and Maasen S, editors.
Human by Nature: Between Biology and the Social Sciences. Erlbaum, Mahwah, NJ.

Jordan P 2014. Technology as Human Social Tradition: Cultural Transmission among Hunter Gatherers. University of California Press, Berkeley.

Shennan S. 2009. Pattern and process in cultural evolution: An introduction. Pages 1-18 in Shennan S, editor. Pattern and Process in Cultural Evolution. University of California Press, Berkeley.

Shennan S. 2011. Property and wealth inequality as cultural niche construction. Philosophical Transactions of the Royal Society B 366:918-926.

Referee: 2

Comments to the Author(s)

I have had the opportunity to read and think about the interesting manuscript entitled, “Cultural linkage: the influence of package transmission on cultural dynamics.” The authors’ introduction highlights the importance of three questions—“how do links between cultural traits emerge? How are they transmitted? And how should the existence of links change our

expectations for the outcome of cultural evolutionary processes?”. They make it clear that they set out to address only the third question in this paper. They employ an agent-based model (the Matlab code is made available but there is no mention of the model description following something akin to the ODD protocol introduced by Grimm et al.) to run simulation experiments designed to inform us of how random linkage between traits might impact cultural evolutionary dynamics in a constant sized finite population of $N=1000$ individuals. They investigate both unbiased and payoff-biased oblique transmission of targeted traits (and those that are linked to the targeted traits). All traits not included in the interaction partner’s cultural “package” are transmitted vertically from the previous timestep. They further assume that the links that bind traits into “packages” can also be transmitted from an interaction partner, though this transmission is imperfect and some links can be broken along the way. They allow for all traits to undergo “innovation” at each time step. They use mean pairwise difference to investigate the effects of the probability of links being broken (b) and the rate at which new links are regenerated (a) on cultural diversity. They investigate hitchhiking under neutral conditions and conditions in which selection against the detrimental associated trait is fairly weak compared to selection for the beneficial target trait. Finally, they assess to what extent equifinality clouds one’s ability to recognize unbiased transmission from cultural diversity if one incorrectly assumes that all traits are independently transmitted (i.e., no links) when in fact they were linked to some extent.

Regarding the ODD protocol, most information required by the protocol is covered in the submitted version of our manuscript, appearing in roughly the same order. We have edited the manuscript to include the missing information. We start with the purpose on lines 85-86, followed by entities, state variables, and a process overview on lines 86-95. Design concepts, along with relevant references are provided as the corresponding part of the model is described (payoff and functionality on lines 97-112, transmission bias on lines 114-122, and innovation on lines 145-149. Some terminology recommended in the ODD protocol is not appropriate for this model. We do not use the terms “adaptation” and “objective” but the corresponding concept “payoff” is described on lines 97-112. Learning, or more generally, cultural transmission, is then described on lines 114-132. The model does not contain prediction and the section is omitted per the ODD suggestion. “Sensing” is described on lines 120-122. The model makes no assumption about whether the interaction is direct, indirect, or facilitated by communication and the section is omitted as allowed by ODD protocol. Stochasticity appears in multiple places in the model and when they appear they are always indicated by the word “probability” or “random” (lines 118, 127, 130, 145-149). There are no collectives in the model and the section is omitted as allowed by the ODD protocol. Observation is now described on line 156-157. There is no input data in the model and we omit this. Finally, there is the Analysis section starting from line 171.

We now cite the ODD protocol on lines 84-85.

Generally speaking, I like the direction of this study. I think the question the authors choose to address is a good one, and I wholeheartedly agree that we should spend more time and energy studying how cultural linkage will affect cultural evolutionary dynamics. I also like the analysis

they have done to address equifinality and the difficulty of identifying individual-level process from population-level pattern. Having said that, my overall impression after reading the study is that the model they present includes more than is needed to address the question they set out to answer. In particular, I do not think the stated goal requires that links between traits are transmitted, broken, and regenerated through the course of a simulation run. For one, it is unclear under what conditions such links are transmitted empirically, at least in humans.

Thank you. As we mentioned in response to a different point from reviewer 1 the actual process by which links are formed between cultural traits and indeed transmitted are not at all clear, as the reviewer correctly points out. The reviewer suggests a model that implicitly assumes a system of psychologically set 'link templates' into which variants are placed, either on a population or an individual level. This may be true in some cases and the effect of static links is an interesting part of the picture. To investigate this, we ran simulations that include a static link structure (see details in Appendix 1). However, more generally, to address the evolution of technology (or the evolution of a tightly linked package of cultural traits) it is crucial that both variants and links between them are transmitted. This allows the package to change in "connectedness" over time and eventually form a group of variants that can be treated as a single unit. This is an important result generated by the original model. However, we thank the reviewer and editor for this suggestion which will enable us to give a more complete picture and pull apart the effect of the presence of links and the effect of their transmission.

But more importantly, it seems the main question posed in this study can be addressed without complicating the model with the transmission of links. I found myself wondering how much of the results are a function of the rather complicated, asymmetrical, and arbitrary rules governing when and how links are broken during transmission relative to how much is a function of the simple presence and number of links (which seems to me to be the main aim of the study).

This is a fair point. The link transmission rules we have implemented are arbitrary. We have tried to ensure that this is clear in the manuscript. We tried where possible to make assumptions based on what is known about the important processes. Where little or nothing is known we have made simple and hopefully sensible assumptions. To examine the effect of our assumptions about link formation and transmission we implement a model in which the links are not transmitted, as suggested by the reviewer. The results are described on lines 222, 246, 261 and Appendix 1.

This ambiguity clouds my understanding of how well the paper addresses the question of interest, which has only to do with the presence of links and not their formation and dynamic change through time. Don't get me wrong, how these cultural packages form and change through time as a function of transmission is an interesting question and one worth tackling. However, it is not the question they set out to address here. In my opinion, including link transmission complicates the task of addressing the question of interest.

We are interested in the effect of links on measures of diversity and the formation of cultural packages. For example, our results show that stable packages have dynamics and signatures very similar to single individual traits. This, we suggest, might be an important piece of the puzzle of the evolution of something like complex technology. However, we agree that the different effects of the presence of links and their transmission should be made explicit. Therefore, we now include simulations where links are not transmitted in Appendix 1 and refer to the results in the main text lines 222, 246 and 261.

I have included a number of line-by-line comments and suggestions below, and I hope they are of some use to the authors. I hope that most of the suggestions are on point, and I apologize if I simply misunderstood or missed something when reading the manuscript—although I have dedicated a good chunk of my time to this, it would always be nice to have more time to spend with reviews.

We very much appreciate the time and energy that the reviewer has given to reviewing this paper as well as the useful suggestions regarding further clarifying simulations.

My most substantive comment/suggestion, however, is that in my opinion the model would be improved by simplifying it even more than it currently is. The goal of this simplification is to address the stated research question more directly and clearly. In particular, I would recommend getting rid of the assumption that links are transmitted. This would allow the authors to jettison some of the parameters (b and a) associated with link breakage and creation. In their place the authors could simply use one parameter (l) to control for the number of (static) links found in each individual throughout the course of each simulation. I would also recommend setting the variable c to 1 instead of 0.99, which means they could go ahead and eliminate c as well.

The model with static links is now included in Appendix 1 and the results are referenced on line 222, 246, and 261 in the main text.

We note that there are two different ways to implement the parameter L . First, it is possible to assign each individual L links between the same traits. This may correspond to some examples of linkage, such as a psychological template that has evolved and exists in all individuals. Here, all linked traits can be viewed as one. Second, it is possible to allow individual variation in the number and location of links, with L being the probability that a link exists at each location. This may correspond to world-views that vary between individuals but are constant within individuals. It is the latter that we focus on in Appendix 1.

The reason we include the variable c is discussed below.

I foresee at least two major upsides to removing the parameters a , b and c along with the assumption that links are dynamic. First, I think the results would be more directly applicable to the question at hand. Second, removing a and b would free them up to run simulations with different values of h and k (at the moment, they only investigate $h=5$ and $k=4$). It would be

interesting to see if the magnitude of the effects of cultural linkage on cultural diversity vary with the total number of traits and the number of variants possible at each trait.

We have included simulations with other values of μ , k , N , c , and h . The results of these are now included in Appendix 3 and referenced in lines 262 and 383. In general, the qualitative results are robust to change in these parameters but we note important differences when the value of c is lower (i.e. when large packages are more difficult to learn). Here, the transmission advantage to a variant in a large package becomes smaller and the effect of variance in package size is diminished.

In sum, at the moment, the model seems more complicated than it needs to be to address the research question. I would recommend using a simpler model to address the stated question in this paper. For the purposes of a second and separate study, they could always modify the model to address how dynamic cultural packages affect cultural diversity if they so desire. It seems to me they have included aspects of the second model that are not needed to address the question they set out to address in this first paper. The model can be pared down to better fit the needs of this project and doing so will likely make the model more general and the results easier to understand in the context of the research question.

line-by-line comments/questions/suggestions

line 10: "that allows links to form and break between cultural traits."

As I discussed above and describe in more detail below, I don't think it is necessary to assume that links break and regenerate to address the central question of this paper. In fact, the study might be improved (made simpler and easier to interpret) if the assumption that links are transmitted is dropped entirely.

Thank you, we have addressed this above and include simulations that do not allow links to be transmitted.

Line 15: There is no mention of the equifinality issue in the abstract even though I think that is one of the more important points to come out of this paper. I'd recommend adding a sentence to the abstract that summarizes that part of the study.

This is a good point. It has been added to the abstract on line 15.

line 18: "Many definitions of complexity rely on some quantification of a trait's component parts and the extent to which those parts are integrated with one another (e.g. [1])."

The authors might benefit from being more precise here. If my memory serves me well, Oswalt focuses on food-getting technology and writes about technounits (the "component parts" in the quote above), but I don't think he would have called a harpoon a "cultural trait." He might use the term "cultural trait" to refer to an individual technounit (though I don't recall him doing

that), but, even then, he makes no distinction between different variants of a technounit. Whether the technounit is made of leather, string, or sinew, it would all be the same to Oswalt as long as the component (the technounit) played the same role in the function of the tool it was a part of. In short, even if one equates Oswalt's technounit with a discrete culturally transmitted component of a tool with a finite number of possible variants (string, sinew, leather)—which is what the authors model in their paper—Oswalt's definition of technounit makes no distinction between those qualitatively different variants. In short, his framework is blind to cultural variation in each technounit. For that reason, Oswalt's work may not be the best example to cite here.

This is a good point and the sentence was not meant to imply that Oswalt's technounit was exactly analogous to our conception of a cultural trait. We hoped to draw attention to the use of the count of constituent parts to describe the complexity of a particular cultural object or artefact. We have replaced the term 'trait' with 'artefact' on line 21 to clarify this and to be more consistent with Oswalt's terminology.

line 21: "This definition suggests that strong links between once-independent components may be a fundamental feature of complex human technology."

Perhaps, or that may be reading too much into the passage. After all, an assemblage of practices does not necessarily require links between them (although I suppose that it often does, as the authors imply).

We interpreted the use of the term 'assemblage' in the definition of technology given by Arthur as implying linkage between independent modules, groups of traits, or individual traits. However, in general an 'assemblage' for example as used in archeology does not necessarily imply co-transmission. However, it is clear from other parts of Arthur's work, for example references to "modularity", that he considers technology to be made up of linked constituent parts in this way (e.g. pp. 35-37). On lines 22 and 24 we now make reference to the page in Arthur's book where this definition occurs as well as parts referring to modularity in technology more generally to draw this out more clearly.

Lines 27-28. "And how should the existence of links change our expectations for the outcome of cultural evolutionary processes?"

Yes, good question. That is the focus of this study.

line 43: "Any cultural trait could be linked to any other producing clusters of traits tightly linked in a network and difficult to separate."

Perhaps I'm taking this too literally, but... Surely there are examples of traits that could not be linked because they are mutually incompatible, no? The details involved in preparing a beef steak dinner probably will not be linked with being vegan. Atheists are unlikely to know which religious ceremonies or prayers to recite on certain days or at certain events. My point is that it

may be too general to state that any trait can be linked to any other; identity is complicated and some traits may not possibly be linked to others depending on the variant expressed at one or the other. Clearly this is not a major critique, but perhaps this sentence can be scaled back to allow for the possibility of non-random links.

This is a great point. We have edited the sentence on line 42 to be clear that we are talking in the most general sense and many counter examples will exist.

line 50: "In the following, for each of these three essential components, we make the simplest possible assumptions to begin to understand the effect of links on cultural evolutionary dynamics."

I don't agree with the claim that "the simplest possible assumptions" are used here. In particular, it is my opinion that the assumption that links are transmitted is not only rather complicated but also unnecessary to address the main question of the study.

We have addressed this in more detail above.

Page 2-3: "Our model assumes that links form at random at a fixed rate between any two traits in an individual's cultural repertoire. This is consistent with a mechanism of link formation where, for example, a role model demonstrates actions in sequence."

The second sentence does not seem to follow from the first. The process described—a model demonstrating something in sequence (say knapping a stone tool)—does not seem like an illustration of how any 2 traits could become linked. In fact, it seems to illustrate a case where the linkage would be quite non-random (if I want to knap a stone tool, I will probably have to copy the traits that are linked in the knapping process exhibited by the model rather than one of the behaviors exhibited as well as the kinship term the role model uses to refer to his mother's brother. Of the other mechanisms listed at the end of this paragraph, perhaps only prestige-biased transmission could conceivably link totally unrelated traits. Even causal reasoning is likely to link traits non-randomly (contrary to Pavlov).

Again, this is a good point and I think a misunderstanding has arisen as a result of our poor phrasing. We have clarified on line 63-64 that we assume the traits have no synergistic effect nor compatibility issues but that this is important future work. We now cite a new study dealing with this on line 50.

Page 3, lines 62-72. This is a very clear paragraph laying out the goals of the paper.

Page 4 "Each individual can form a link between any two traits, ..."
With what probability? When?

We now include a reference on line 93 to the section below where this information is detailed.

Page 4, lines 86 – 101. I take it an individual's "payoff" (Eq 2) is meant to be synonymous with its "fitness" or perhaps cultural influence and that $s_i > 0$ is meant to represent the strength of (cultural) selection acting on different variants of the trait, but I'm not 100% sure if that is what the authors mean. If so, it might be worth making that connection explicit with a sentence or two so as not to leave the reader guessing about the terminology. If not, then some text should be added that explains the distinction.

Under payoff-biased transmission the payoff has the same effect as fitness. However, we did not use the word "fitness" to avoid ambiguity, especially because there is no true birth death process in the model. We have added empirical examples of payoff on line 99 to make the term easier to grasp before we explain how individuals chose the interaction partner.

Page 5, line 107: I assume the payoff-based interaction does not allow an individual to pair with itself. Is that the case?

The reviewer is correct. In all cases individuals choose interaction partners from the population apart from itself. We have clarified this on line 116.

line 109: "Multiple individuals may choose the same interaction partner."

This makes sense, but it raises a related question that I do not see the answer to in the text: how/when do individuals update their variant value at each transmitted trait? Say Bob is chosen as the interaction partner by two others: Sally and Tate. Bob's interaction partner is Frieda, from whom he tries to copy trait 1. Further assume that Sally and Tate both attempt to copy Bob's variant at trait 1. However, Sally attempts to copy Bob before Bob copies trait 1 from Frieda, while Tate attempts to copy Bob after Bob's interaction with Frieda. Assuming the variant Bob acquired from Frieda was different from his previous variant at trait 1, Sally and Tate would have copied different variants from Bob if individuals update variant values in real time (i.e., asynchronously). Alternatively, if all individuals update synchronously, then Sally and Tate would have copied the same variant value from Bob, because Bob would not have updated his variant at trait 1 (the one he got from Frieda after teaching Sally but before teaching Tate) until the end of the time step. In my opinion, synchronous updating (i.e. everyone in generation t attempts to copy the variants displayed by their interaction partner in generation $t-1$) would make the results easier to interpret, but I can't tell what was done in the case of this model. Please note that I did not look at the code to find out, though of course the answer is there. Perhaps it is in the text and I just missed it somehow, but the reader should be get this info without looking into the code.

This is a fair point, too. The updating is synchronous and the text now reflects this on line 87.

Page 5: "Each variant is copied successfully with probability c , while unsuccessful copying events mean that the focal individual keeps its original variant."

I admit that I find this decision a bit confusing. If I understand correctly, $1-c$ does not represent

copying error, per se. That is, $1-c$ is not meant to be the cultural analog of mutation. Instead, c simply represents the probability that a given trait (the chosen one or one linked to the chosen one) will get transmitted from the interaction partner to the focal individual. I'm not sure what its utility is here. More specifically, why would c ever be less than 1? And if c doesn't represent copying error, then, for better or for worse, copying error is missing from this model, which might be an oversight. If $c=0$, nothing gets transmitted obliquely and the model defaults to vertical transmission of unlinked traits. So, all in all, I take it that c represents the strength of oblique transmission relative to vertical transmission of the chosen trait and those that are linked to the target trait. If that is the case, perhaps it is best to explain it that way (or perhaps I misunderstood the text). To isolate the effects of linkage, I would simply set c to 1 for the purposes of this paper, which would mean that I would not include c at all. Removing c (or setting c equal to 1) would mean that all of the traits linked to the target trait would be passed via oblique transmission and all traits outside of this "package" would be passed via vertical transmission. Wouldn't this also make for a cleaner comparison to the W-F expectations under conditions in which oblique transmission is unbiased and all 5 traits are linked to each other? The value of c used in the paper is very close to 1 (0.99), but I do not recall an explanation for that value or for why it is not varied in the experimental design. I think I would simply get rid of c , which is the same as setting it to 1 instead of 0.99.

We included c to account for the possibility that large packages are harder to learn. Including c , makes packages harder to transmit in full as package size increases. This is now stated on line 128-129.

Page 5 "the focal individual acquires each link in the package with probability $1-b$, so that with probability b a link in the transmitted package is broken."

I continue to question the decision to make links transmittable. I question this decision not because the assumption is "unrealistic," but because it seems to make for a weaker experimental design. Of course, the package of variants needs to be transmitted obliquely—that much is crystal clear. But I do not think the links need to be transmitted to address the following question: "And how should the existence of links change our expectations for the outcome of cultural evolutionary processes? In this paper we focus on the last question and present a model where cultural traits can be linked together and transmitted as a package."

To my mind, there is no reason to make the links transmittable to address that question. Dropping that notion would also allow you to get rid of the parameter, b . It would also do away with the asymmetry in how the transmission of links is dealt with: at the moment b applies only to links that are present in the interaction partner and not to the links that the focal individual has but the partner does not—the latter are essentially overwritten by the links present in the partner. Ego's links that connect to traits outside of its interaction partner's linked constellation are also erased. This is why the number of links drops to 0 when $a=0$. If one does not need to worry about transmitting links (which is not really the stated purpose of this paper and actually seems to be a more difficult and involved question than what is being addressed here), then one does not need b at all. And if links cannot be broken because they

are not transmitted, then there is no need for a . It seems to me the experimental design would be much simpler if you do not worry about the transmission of links. You could introduce a new parameter, l , to represent the number of links in each individual. Those links would be constant in each individual throughout each simulation run. You could vary l between 0 and 10 to arrive at the same conditions illustrated in Figure 2 without bothering with a and b at all. In short, it strikes me that while including the transmission of links introduces the need for a number of variables and tough decisions about how links can be passed and broken, it does not lead to insights great enough to outweigh the costs. For these reasons, I'd advise against modeling the links as transmittable. This results in a model with fewer moving parts, fewer parameters, and fewer assumptions but clearer results for addressing the question of interest. Save the transmission of links for a second step.

This is now addressed in Appendix 1, where we describe simulations that do not allow for link transmission.

line 132: "After transmission, new links randomly form at the rate of association, a between any of an individual's traits that are not linked."

Again, I'd get rid of the notion that links are transmitted. It does not seem necessary to address the stated question. If links are not transmitted, they cannot be broken. If they cannot be broken, then there is no need to regenerate them through this process involving a . Each individual will have the same links throughout the simulation.

This has been addressed above.

line 133: "Individuals may then innovate a variant for each trait with probability μ ."

So, this model represents "innovation" rather than copying error. I'm concerned that "innovation" might connote progress or guided variation, while the algorithm provided shows that any potential variant value is equally likely to be adopted as a result of "innovation" during this process. Is this the case?

Yes, this is correct.

I think I would call this "copying error" so as not to imply that "innovations" are directed to better values (in fact, in the case of the neutral traits, none of the variant values are more fit than any other). Because some of the traits are obtained obliquely (those copied from the interaction partner) and some are obtained vertically (those not copied from the interaction partner but copied from ego's previous generation), the term copying error is still legit in this case, I think.

We use the term 'innovation' as defined by Cavalli-Sforza and Feldman (1981). Here, they mean 'innovation' to be directly analogous to genetic mutation. They provide a discussion of this definition in relation to innovation as a 'directed' phenomenon on p. 66. We now say this

and provide a reference to this where we introduce innovation on line (146-148) to make our intended meaning clear.

Why was only one innovation rate tested? Why not more? If you get rid of parameters a, b, and c, it might be interesting to test additional values of μ , h, and k in your experimental design. I would be very interested to see how the results vary with greater h and k, for instance.

We now include simulation results with different μ , h, k, c, and N in lines 383 and Appendix 3.

lines 145-150: I see why those “adjustments” are needed to address the hitchhiking question, but they are so heavy-handed that they raise the question of why even conduct the burn-in period in the first place? It seems like the population can be set as needed in this case without the burn-in period.

We agree that the reason for these adjustments is not intuitive and we have tried to make it as clear as possible in the text. We have edited the text (now line 158-166) to further clarify the need for these adjustments by moving our rationale as well as the effect of the adjustments to the start of the paragraph. The burn-in period is necessary to allow the frequency of links to stabilize, while the adjustments make the effect of hitchhiking more obvious when it occurs. We do not make any claims about the validity of our starting conditions here – the simulations only serve to prove that hitchhiking can occur and its signature could be mistaken for a signature of selection.

lines 169-170: “In contrast, if π_i has a value close to 1 then the variants occur with almost equal frequency in the population.”

I think it might be worth modifying this slightly to read something like “higher values represent populations in which all variants occur with equal frequency” because just how closely the value approaches 1 when all variants are represented equally is dependent upon the number of variants possible at the trait. For example, for the conditions of this model, if each of the 4 variants of a trait appeared in exactly 250 of the N=1000 individuals, the resulting value would be 0.751. 0.751 is obviously closer to 1 than 0 is, but one might not consider 0.751 “close to 1” in an absolute sense even though all 4 variants are represented equally in the population. Now imagine there were 100 possible variants and each was displayed by 10 of the N=1000 individuals. Now the value (0.991) is indeed close to one in the absolute sense even though the variants are just as evenly represented among the 100 variants as in the previous case with 4 variants. If only 2 possible variants are distributed equally, the value isn’t very close to 1 in the absolute sense.

We have changed the wording. The sentence (line 179) now reads “if π_i is high the variants...”

line 195: Is Figure 2 depicting the mean pairwise difference of just one trait or the mean of the mean pairwise differences of all 5 traits? This is not specified in the caption.

It is the mean pairwise distance at just one trait. This information has been added to the caption.

line 206: “When $n=0$ there is only innovation and no transmission, and so the pairwise difference is maximized.”

This might be a semantic quibble, but isn't there still vertical transmission in this case instead of no transmission? $n=0$ simply means none of the traits were obtained via oblique transmission and all traits were obtained via vertical transmission from the previous timestep.

We interpret the model generations as successive horizontal transmission events. There is no demography in the model, so individuals do not die or reproduce and there is no vertical transmission. We now state this clearly on line 88.

lines 210-212: “To do this, we scale the rate of innovation by h/nc to account for the fact that innovation occurs at a constant rate in each timestep while the traits are, on average, only successfully transmitted every h/nc timesteps.”

This seems like an example of where the analysis would be simpler if the authors eliminated the variable c altogether.

The reason for including c has been addressed above.

lines 227-240: This is an interesting paragraph. It makes sense that variable package size would also have an interesting effect on the transmission of some traits (those that are located within a large cluster) over others (those located in a smaller cluster). If the authors were to take my advice and keep the number of links constant throughout each simulation by dropping the notion that links are transmitted (and thus dropping the need for a and b), then within each simulation there would probably be less variance in package size—e.g., 4 links could connect either 4 traits or 5 traits—than what they found in their simulations so far. Simplifying their model in this way might reduce the magnitude of the effect of variance in trait cluster size they describe here. That would be a cost, but the benefits of simplifying the model might outweigh this cost.

The reviewer is correct that static, non-transmitted links decreases the magnitude of the effect shown in Figure 2 considerably by removing the transmission advantage of variants in large packages over successive learning events. This is shown in Figure A1.2. More details are available in Appendix 1.

lines 231-233: “In Fig. 3B we show the variance of the package size – where the variance is high

the benefit of being in a large package is highest. The variance becomes lower when a is high or b is low because in these cases all traits are linked in large packages.”

But it looks to me that the variance can also be low (note: the y-axis of fig. 3b should be flipped so the values increase as they go up) when a is low. And the results show that variance can still be quite high when b is assigned the lowest value tested here. In short, the description provided in the text seems incomplete, if not contradicted, by the results presented in fig. 3b.

Thank you for pointing this out. This was an incomplete description of the figure. Variance can be low when all traits are linked (i.e. when a is high and b is low) or when no traits are linked (i.e. when a is low). The realised variance in package size in any particular simulation is, therefore, a complex interaction between a and b as shown in Figure 3B. This is now more clearly explained on lines 240-241.

lines 238-240: “This suggests an important interplay between package size and package stability over time in driving cultural evolution with linked traits.”

Yes, this is an intuitive but interesting and important point worth highlighting.

lines 241-256: This is a very good point about equifinality. I wonder if the results would be similar in the simpler model I’ve suggested in which links are not transmitted and a, b, and c are jettisoned? If not, then that would be a bit concerning.

The results are similar. The area of overlap between the model in which links are not transmitted and the Wright-Fisher model is 0.87, and between that model and an unlinked model is 0.45. This indicates that this issue with equifinality can exist in the non-transmitted links model, as in the original model. As with the original model, the overlaps depend critically on the parameter values. Details can be found in Appendix 1 and in Figure A1.3.

line 259: “conformist-biased transmission”

Where did talk of conformist transmission come from? This has not been discussed at all in the paper to this point. It seems a bit jarring to throw it in at this point.

We have added description of conformity in the abstract, lines 116, 119-120, and 155-156.

line 265: “See Appendix, Section 2 for how conformity is modelled.”

I would argue in this case that one should not have to go to the appendix to see how conformity is modelled. If it is important enough to be included in one of the main figures in the text, it should probably be explained in the main text along with the other major components of the model.

We now explain how conformity is modelled in lines 116, 119-120, and 155-156.

line 284: “We have shown that in situations of low (but positive) or intermediate link frequency the level of cultural diversity as measured by the pairwise difference is lower compared to the Wright-Fisher expectation (cf. Figs. 2A,B).”

Have I misunderstood something, because Fig. 2B shows that this is not the case for $b=0.1$? When $b=0.1$, the pairwise difference remains higher than the W-F expectation when a is low (but positive). As a related side note, it would be nice to see the error bars around the means presented in Fig. 2.

When $b=0.5$, the pairwise difference remains higher than the W-F expectation for all values of a . It remains higher because when $b=0.5$, packages are broken down at a relatively rapid rate. We have changed the sentence to read “...the pairwise difference *can be* lower than the Wright-Fisher expectation.”

We did not include error bars because they make the figure difficult to interpret. We now point to Figure 4 in the figure legend. This shows the entire distribution of pairwise difference for a specific set of a and b values and gives an indication of the variation around the means presented in Figure 2.

line 314: “We show only estimates of this mean time for parameter constellations where hitchhiking occurred in sufficiently many simulations (at least 30) and...”

This sounds vague and somewhat arbitrary. What makes 30 “sufficiently many,” and out of how many?

This is an arbitrary cutoff point. It was set because we wanted to avoid presenting unreliable and noisy results that were supported by a very low number of simulations. To give an idea of the amount of data included we now include the cutoff point as a percentage of total simulation runs on line 315.

line 375: “Further, hitchhiking becomes less likely the more detrimental the associated variant.”

OK, but does it still happen to some degree as long as $s_1 > s_2$? Or does the fact that the links are broken frequently in the current model mean that hitchhiking won't occur at all if the associated trait is detrimental?

When the hitchhiking variant is detrimental, hitchhiking can still occur but the probability of hitchhiking occurring becomes lower. Where links are broken very frequently, hitchhiking is also much less likely to occur (compare, for example, lines for $b=0.5$ and $b=0.1$ in Figure 5 panel D). So, although hitchhiking can occur where traits are detrimental, the probability will depend on the values of s_1 , s_2 and link frequency.

This is another place where allowing links to be passed, broken, and created during the simulation may make things more complicated than they need to be this early in the game.

Where links are transmitted both the beneficial and hitchhiking traits and the links between them can be transmitted. This maintains a relationship between the traits and facilitates hitchhiking.

Where links cannot be transmitted, the traits may be passed on but the link between them depends on the link structure of the receiver. This makes hitchhiking less likely as shown in Figure A1.4.

line 384: “We note that by design the modelling assumptions are as simple as they can be to begin to understand the effect of links on cultural evolution...”

I disagree with “as simple as they can be”; the model can be simplified further by getting rid of the idea that links are transmitted according to what turns out to be a rather complicated set of rules. I am of the opinion that the costs of including link transmission outweigh the benefit at this early stage.

line 386: “We assume that links form at random at a fixed rate between any two traits in an individual’s cultural repertoire and can be broken through the process of cultural transmission.”

My major suggestion for this study is to do away with those assumptions altogether; to start simpler with this first model on the topic. It might be worth returning to the issue of how dynamic link formation might influence cultural diversity, but that seems a separate and more difficult question than the stated focus of this paper.

line 388: “Whether this assumption is appropriate depends on the study system and the nature of specific links.”

Fair enough, but I’d also argue that this assumption may not even be appropriate for the question at hand, regardless of any actual study system one might be interested in down the road.

As mentioned above, we now examine a model with static, non-transmitted links. The results draw out interesting differences between the effect of the presence of links and their transmission.

line 397: “The effects of links on pairwise difference interact with the rate of innovation: the smaller the rate, the lower the pairwise difference for low or intermediate link frequencies (see Appendix, Section1, Fig. A1.1A,B)”

Is this point important enough to include the figure that supports it in the main paper rather than in the appendix? I think it might be.

The original statement in the manuscript was incomplete and we have rectified this. As shown in Appendix 1, the patterns generated by different values of μ , k , h , and N are qualitatively the same relative to the appropriate WF expectation, although they differ quantitatively. It is this quantitative difference that we were hoping to draw attention to in the original statement. We now make this clearer and more accurate on line 383. Figures are provided in Appendix 3 due to space constraints.

line 418: "In our model the linkage patterns can vary between individuals and are copied along with trait variants, allowing variants in large packages to spread faster than those in smaller ones."

As described above, I am not sold on the need to allow for links to be transmitted in the current version of the model. I think there would still be a way to address the issue of how a larger number of links can enhance the speed with which a trait in a larger package can spread versus a trait in a smaller package by simply controlling for the number of links found in all individuals at the start of the simulation (e.g., all individuals in a run would have 3 links and they would keep those for the entire simulation regardless of their interaction partners' links) as an experimental parameter called "l" for links, but disallowing those links to be passed, broken, or regenerated through the course of a simulation. I may be wrong, but to my mind, that would be a simpler and cleaner way to address the central issue of this paper.

In Appendix 1 we provide a model where links are not transmitted but do vary between individuals.

lines 438-442: These are some very interesting thoughts for future work.

Thank you!